# Asthma-associated genetic variants induce *IL33* differential expression through an enhancer-blocking regulatory region

Ivy Aneas [1,7 ✉], Donna C. Decker[2,7], Chanie L. Howard[3], Débora R. Sobreira [1], Noboru J. Sakabe[1], Kelly M. Blaine[2], Michelle M. Stein [1], Cara L. Hrusch[2], Lindsey E. Montefiori[1], Juan Tena [4], Kevin M. Magnaye [1], Selene M. Clay [1], James E. Gern [5], Daniel J. Jackson[5], Matthew C. Altman [6], Edward T. Naureckas[2], Douglas K. Hogarth[2], Steven R. White[2], Jose Luis Gomez-Skarmeta[4], Nathan Schoetler [2], Carole Ober[1], Anne I. Sperling[2,3,7 ✉] & Marcelo A. Nóbrega [1,7 ✉]

Genome-wide association studies (GWAS) have implicated the *IL33* locus in asthma, but the underlying mechanisms remain unclear. Here, we identify a 5 kb region within the GWAS-defined segment that acts as an enhancer-blocking element in vivo and in vitro. Chromatin conformation capture showed that this 5 kb region loops to the *IL33* promoter, potentially regulating its expression. We show that the asthma-associated single nucleotide polymorphism (SNP) rs1888909, located within the 5 kb region, is associated with *IL33* gene expression in human airway epithelial cells and IL-33 protein expression in human plasma, potentially through differential binding of OCT-1 (POU2F1) to the asthma-risk allele. Our data demonstrate that asthma-associated variants at the *IL33* locus mediate allele-specific regulatory activity and *IL33* expression, providing a mechanism through which a regulatory SNP contributes to genetic risk of asthma.

[1] Department of Human Genetics, University of Chicago, Chicago, IL 60637, USA. [2] Department of Medicine, Section of Pulmonary and Critical Care Medicine, University of Chicago, Chicago, IL 60637, USA. [3] Committee on Immunology, University of Chicago, Chicago, IL 60637, USA. [4] Centro Andaluz de Biología del Desarrollo (CSIC/UPO/JA), Universidad Pablo de Olavide, Seville 41013, Spain. [5] Department of Pediatrics, University of Wisconsin School of Medicine and Public Health, Madison, WI 53726, USA. [6] Division of Allergy and Infectious Diseases, Department of Medicine, University of Washington, Seattle, WA 98195, USA. [7] These authors contributed equally: Ivy Aneas, Donna C. Decker, Anne I. Sperling, Marcelo A. Nóbrega. ✉email: ianeas@bsd.uchicago.edu; asperlin@uchicago.edu; nobrega@uchicago.edu

Asthma is a common and chronic inflammatory disease of the airways, with significant contributions from both genetic and environmental factors. Genetic factors account for more than half of the overall disease liability[1] and genome-wide association studies (GWAS) have discovered more than 60 loci contributing to asthma disease risk[2], with most of the associated variants located in noncoding regions. Linking these noncoding variants to genes and understanding the mechanisms through which they impart disease risk remains an outstanding task for nearly all asthma GWAS loci.

Among the most highly replicated asthma loci are variants near the genes encoding the cytokine IL-33 on chromosome 9p24.1 and its receptor, ST2 (encoded by *IL1RL1*), on chromosome 2q12.1, highlighting the potential importance of this pathway in the genetic etiology of asthma. A crucial function for IL-33 in allergic airway inflammation and bronchial airway hyperresponsiveness has been known since its discovery in 2005[3]. Studies in individuals with asthma and in mouse asthma models have identified elevated levels of IL-33 protein in both sera and tissues[4,5]. This cytokine is a potent inducer of type 2 immune responses through its receptor ST2 and has been broadly implicated in other allergic and inflammatory conditions, such as atopic dermatitis, allergic rhinitis, and eosinophilic esophagitis[6–8].

The single-nucleotide polymorphisms (SNPs) associated with increased asthma risk at the *IL33* GWAS locus reside within a linkage disequilibrium (LD) block in a noncoding genomic segment located 2.3 kb upstream of the *IL33* gene. We, therefore, posited that variants in this region impact on *IL33* expression by altering *cis*-regulatory element(s) that control quantitative, spatial and/or temporal-specific gene expression. Previous studies of complex diseases have shown how regulatory variants in promoters and enhancer elements lead to an increased risk of disease through altering the expression of nearby genes[9–13]. In contrast, other types of *cis*-regulatory elements, including repressors and insulators (also known as enhancer-blocking elements), are less understood and characterized than enhancers, but are also likely to be functionally modified by regulatory variants[14].

Here, we combine genetic fine-mapping using GWAS datasets with functional annotations from relevant tissues to characterize the asthma-associated region upstream of the *IL33* gene. We identify a regulatory element containing SNPs that control *IL33* expression. Genotypes at rs1888909, a SNP within this regulatory element, are associated with *IL33* expression in ethnically diverse populations, as well as IL-33 plasma protein levels. Our study provides functional insights into the function of common regulatory variants at the *IL33* locus and illustrates how a causal SNP can exert phenotypic effects by modulating the function of regulatory elements that do not fit into standard definitions of enhancers, insulators, or repressors.

## Results

**Defining the *IL33* locus asthma-associated critical region.**
Variants at the *IL33* locus have been robustly associated with asthma in GWAS of ethnically diverse populations[2,15–20]. We first used LD between the most significantly associated SNP in each GWAS (here on referred to as the lead SNP) and other SNPs to define the region harboring potentially causal variants at this locus (Fig. 1a). Five lead SNPs were reported among seven large GWAS, defining an LD block spanning 41 kb in European ancestry individuals (chr9: 6,172,380–6,213,468, hg19; Supplementary Fig. 1a). Because LD tracts are longer in European genomes compared to African ancestry genomes, we also sought results of GWAS in African Americans to potentially narrow this region. Three multi-ancestry GWAS[16,18,19] included African Americans, but only one[16] provided GWAS results separately by

ancestry. In that study, the lead GWAS SNP rs1888909 differed from the lead SNPs in the European ancestry (rs1342326[15], rs928413[17]), rs7848215[20], and rs992969[2]) or combined multi-ancestry (rs2381416[16]) and rs992969[18,19] GWAS. Because there was so little LD in this region in African Americans we used an $r^2 \geq 0.40$ to define LD. SNPs in LD with rs1888909 at $r^2 \geq 0.40$ in African Americans defined a region of 20 kb (chr9: 6,188,124–6,209,099, hg19) that excluded two lead SNPs, rs928413[17] and rs7848215[20] (Supplementary Fig. 1b and Supplementary Table 1).

Next, we used Roadmap Epigenome[21] and ENCODE[22] data to annotate the regulatory landscape of the more inclusive 20 kb asthma-associated interval defined by LD. This segment was enriched with chromatin marks and DNase hypersensitive sites (Fig. 1b), suggestive of regulatory potential in multiple cell types. We also identified two CTCF sites within 2 kb of each other with evidence of CTCF binding in multiple cell lines (84 cell lines have CTCF binding to site 1 and 142 have CTCF binding to site 2 out of 194 ENCODE-3 lines). In addition, the cohesin complex RAD21-SMC-3 subunits and zinc-finger proteins such as ZNF384 and ZNF143 also bind to this region (Fig. 1c). Binding of this multi-subunit complex along with CTCF can provide sequence specificity for chromatin looping to promoters or have insulator functions[23,24].

Interestingly, the three lead GWAS SNPs (rs1342326[15], rs2381416[16], and rs992969[2,18,19]) in the LD-defined 20 kb region did not overlap with regions of open chromatin or transcription factor binding; one, rs992969[2,18,19], mapped within heterochromatin (thick blue horizontal bar) in virtually every Epigenome Roadmap cell type (Supplementary Fig. 2). For these reasons, it is not likely that any of these three lead SNPs within the LD-defined region are causative of the asthma association. In contrast, five SNPs in high LD with the asthma-associated lead SNPs overlapped with a region of open chromatin and CTCF, cohesin and ZNF binding, delineating a discrete 5 kb region (chr9: 6,194,500–6,199,500, hg19) (Fig. 1b–d).

If the 5 kb region acts as a regulatory element of *IL33*, it is expected that it would form long-range chromatin interactions with the *IL33* promoters. To investigate this, we analyzed data from promoter capture HiC (PCHi-C) in bronchial epithelial cells harvested from lungs[25] and human lymphoblast cells[26]. We found that the 5 kb asthma-associated region is able to physically interact with the *IL33* promoter in both cell types (Supplementary Fig. 3a). We further confirmed this interaction by analyzing virtual 4C data constructed from Hi-C data of K562 and human umbilical vein endothelial cells[27] using the 3D genome browser (http://3dgenome.org/) (Supplementary Fig. 3b).

Collectively, these data characterize a 5 kb region that harbors both asthma-associated SNPs and marks of regulatory activity that could modulate *IL33* expression through long-range chromatin interactions.

**Regulatory properties of a 5 kb region upstream of *IL33*.**
Having identified a region of interest, we sought to examine its impact on *IL33* expression. Because *IL33* is expressed in multiple tissues, we used an in vivo model system to assay for the spatial regulatory properties of this 5 kb region. However, the lack of evolutionary conservation at the locus between human and mouse (Fig. 1d) required the creation of a "humanized" transgenic mouse model. For this, we used a human Bacterial Artificial Chromosome (BAC; clone RP11-725F15), approximately 166 kb long, spanning the coding region of the *IL33* gene and its upstream sequences, including the 20 kb asthma-associated region. We recombined a cassette containing a E2-Crimson reporter with a 3′ stop codon into exon 2, in frame with the *IL33*

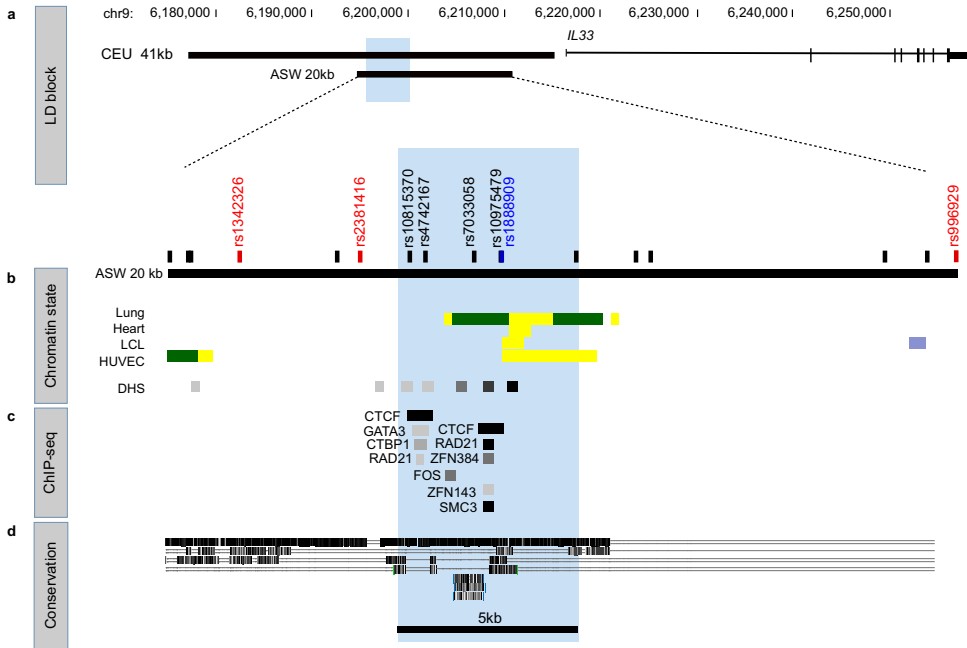

**Fig. 1 Epigenetic characterization of the asthma-associated critical region in the IL33 locus. a** Schematic organization of the *IL33* gene and the asthma-associated region (black bars) of European ancestry (CEU 41 kb, chr9: 6,172,380–6,213,468; hg19) and African ancestry (ASW 20 kb, chr9: 6,188,124–6,209,099; hg19) positioned upstream of exon 1. **b** Position of the lead GWAS SNPs (in red) and additional SNPs in high LD ($r^2 \geq 0.8$) with the lead SNPs (in black) within the ASW 20 kb LD region. The lead SNP rs1888909 in African ancestry is shown in blue. Chromatin states from Roadmap Epigenomics Project showing regions with potential regulatory activity. Yellow: active enhancer; green: transcribed sequence; blue: heterochromatin. DNase hypersensitive (DHS) sites indicating open chromatin regions are shown. Tissues (from the top): E096 Lung primary HMM; E095 Left ventricle primary HMM; E116 GM128781 Lymphoblastoid cell primary HMM; E122 HUVEC Umbilical Vein Endothelial Primary Cells Primary HMM. **c** ChIP-seq data from ENCODE-3 cell lines (338 factors; 130 cell types) showing co-binding of CTCF, RAD2, ZFNs, and SMC-3 at the 5 kb interval (blue shaded region; chr9: 6,194,500–6,199,500; hg19). **d** UCSC Multispecies conservation showing that the 5 kb region is not conserved across species. Organism (from the top): rhesus; mouse; dog; elephant. A single line means no bases in the alignment, and double lines indicate one or more unaligned bases.

start codon (ATG). Any *IL33* regulatory regions within this BAC would drive E2-Crimson expression, mimicking *IL33* endogenous spatio-temporal expression patterns. Also, to directly test the regulatory impact of the 5 kb region, we selectively deleted this asthma-associated DNA segment from the full BAC and assessed the resultant *IL33* expression in vivo, in mice harboring either the full length or the 5 kb deletion BAC (Fig. 2a). The estimated BAC copy number varied from 2 to 11 between founders. Copy number and Crimson expression for each line containing either the full BAC ($n = 5$) or the 5 kb deletion ($n = 4$) are shown in Supplementary Fig. 4.

Immunofluorescence staining of peripheral lymph node and trachea from full BAC transgenic mice (h*IL33*^Crim^BAC) showed that E2-Crimson fluorescent protein is highly expressed in these tissues (Fig. 2b and Supplementary Fig. 4a). Strikingly, constitutive expression of the E2-Crimson reporter was co-localized with the endothelial cell marker CD31 and observed in high endothelial venule (HEV) cells in mouse lymph nodes. This observation validates the species-specificity of the E2-Crimson expression, as previous studies report that while IL-33 is produced by HEV in humans, it is not found in mouse HEV[28]. *IL33* is also highly expressed in basal epithelial cells in the lungs. Unlike humans that have basal epithelial cells throughout the lungs, mouse basal cells are primarily limited to the epithelium in the trachea. The mouse basal epithelial cells from the h*IL33*^Crim^BAC transgenic mice express the Crimson reporter demonstrating a similar expression pattern to that found in human basal cells.

The deletion of the 5 kb region in the reporter BAC (h*IL33*^Crim^BAC5kdel) significantly depleted E2-Crimson immunostaining in lymph node HEV and tracheal basal epithelial cells

compared to the staining observed in the full BAC mice (Fig. 2c and Supplementary Fig. 4b). The 5 kb deletion also significantly reduced E2-Crimson mRNA expression in heart and lung in three out of four independent lines (Fig. 2d). These results show that the BAC encodes human-specific regulatory patterns in vivo and demonstrate the importance of the 5 kb noncoding segment for proper *IL33* expression.

**Asthma-associated SNPs modify regulatory properties of the 5 kb region**. To functionally characterize the regulatory impact of allelic variants of asthma-associated SNPs in the 5 kb region, we utilized a combination of in vitro and in vivo reporter assays. Because this region overlaps with chromatin states suggestive of enhancer function in some cell types (Fig. 1b), and because its deletion in the BAC resulted in a generalized loss of reporter expression in vivo, we first hypothesized that the 5 kb region corresponds to an enhancer. To test the regulatory potential of this DNA segment in vitro, we cloned both the risk and non-risk haplotypes of the 5 kb fragment into a luciferase vector and transfected them in human cell lines. We used an immortalized human aortic endothelial cell line (TeloHAEC) and K562, as the ENCODE project chromatin data annotate this region as a putative enhancer in endothelial and blood cells. We failed to detect enhancer activity of either haplotype in the 5 kb fragment (Supplementary Fig. 5).

Interestingly, upon close inspection of ENCODE data we notice that this 5 kb region is not bound by transcription factors usually associated with enhancer activity, including transcriptional activators, repressors, or RNA Pol II. Rather, it is bound by both CTCF and subunits of the cohesin complex across multiple cell lines (Fig. 1c).

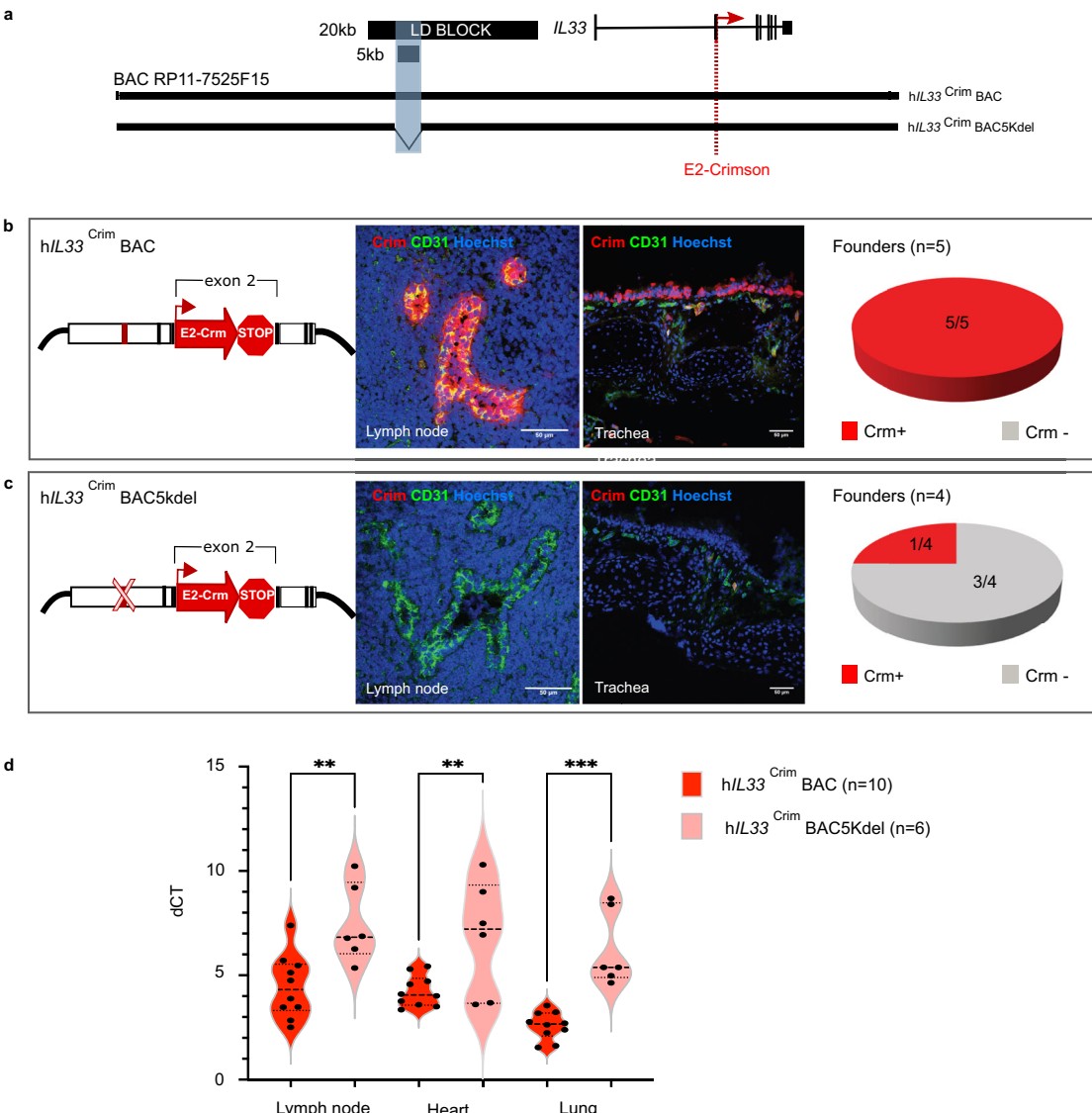

**Fig. 2 The IL33-containing BAC in transgenic mice encodes human-specific regulatory patterns and demonstrates the importance of the 5 kb noncoding segment for proper IL33 expression. a** Schematic of human BAC clone RP11-725F15 (166 kb) spanning the entire coding region of *IL33* and its upstream region including the 20 kb asthma-associated interval and the 5 kb region of interest shaded in blue (black bars). To produce a human *IL33* reporter strain, a cassette containing E2-Crimson with a stop sequence was inserted into exon 2, in frame with the *IL33* translational start site (red dotted line). Transgenic mice were generated with either the full BAC (h*IL33*^CrimBAC) or a BAC containing a deletion of the 5 kb interval within the LD block (h*IL33*^Crim BAC5kdel). **b**, **c** Immunofluorescence staining of mouse peripheral lymph node sections (left panels) and trachea tissue sections (right panels) of E2-Crimson in h*IL33*^Crim BAC mice (**b**) or h*IL33*^Crim BAC5kdel (**c**). Representative founder BAC transgenic lines are shown. Sections were stained with anti-E2-Crimson (red) and the mouse endothelial cell marker CD31 (green). Hoechst 33342 staining for nuclei is in blue. Pie charts show the distribution of Crimson expression (Crm) in each "humanized" BAC mouse line. **d** qPCR analysis of E2-Crimson mRNA obtained from lymph node, heart, and lung from both BAC strains is shown. Violin plot shows average dCT values (Crimson/Ppia) obtained from animals containing either the full BAC or the 5 kb deletion ($n = 10$ biologically independent animals for full BAC and $n = 6$ biologically independent animals for 5 kb deletion). Center line: median; box limits: upper and lower quartiles; whiskers: 1.5 × interquartile range. **p = 0.0017 (lymph node); **p = 0.0066 (heart), ***0.0001; one-way ANOVA with post hoc Sidak multiple comparison test. Source data are provided as a Source Data file.

Both CTCF and cohesin are key determinants of chromatin loop formation and stabilization, including the positioning of regulatory elements close to the promoters of their target genes, or serving as insulators, with enhancer-blocking properties.

To test the enhancer-blocking properties of the 5 kb fragment, we first used an in vitro luciferase-based assay in which candidate sequences containing the risk and non-risk haplotypes are cloned between a strong promoter (SV40) and a strong enhancer (HS2 element in human beta-globin LCR)[29]. As a control, we used a

5 kb DNA sequence devoid of any epigenetic marks of active chromatin and with no evidence of CTCF or cohesin binding (Fig. 3a). Luciferase expression is driven by the enhancer and promoter elements, and a decrease in luciferase activity would be interpreted as enhancer-blocking activity, with the enhancer not able to loop to the adjacent SV40 promoter. When the 5 kb non-risk region was cloned into this vector we observed prominent enhancer-blocking activity compared to a same size control sequence (Fig. 3b). Significant differences were also observed

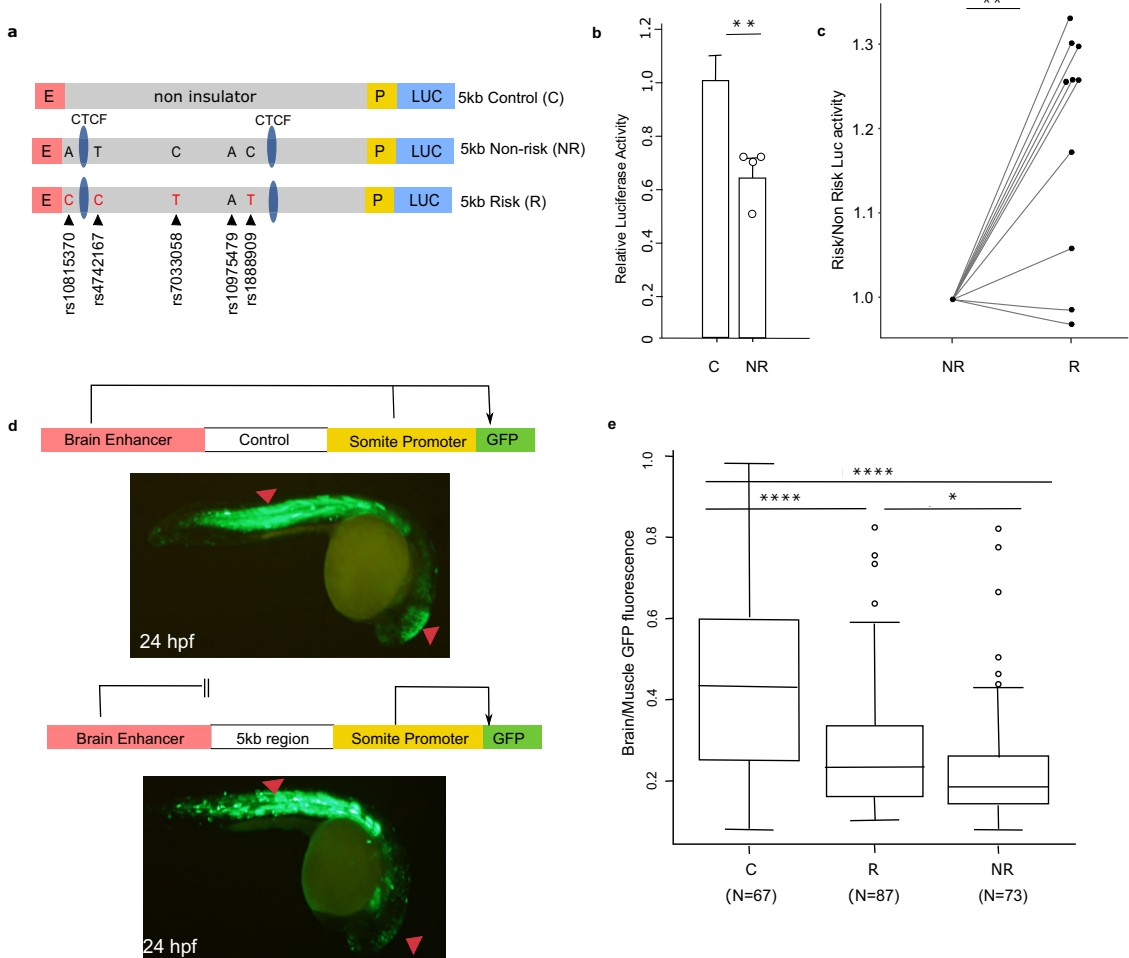

**Fig. 3 Impact of the asthma-associated variants in the regulatory property of the 5 kb region. a** In vitro transgenic reporter assay. Luciferase-based enhancer barrier assay using 5 kb constructs (non-risk or risk; chr9: 6,194,675–6,199,500; hg19) that were cloned between HS2 enhancer (E) and SV40 promoter (P) sequences and transfected into K562 cells. SNPs in the construct are noted (black arrowheads). **b** Enhancer barrier activity of the non-risk 5 kb region of interest compared to a same size control insert (chr7: chr7: 35,303,890–35,309,030; hg19). Data are presented as mean ± SEM, $n = 4$ independent experiments. **$p = 0.0063$, one-tailed paired Student's $t$-test. **c** Luciferase activity values of the 5 kb risk construct is shown as fold change over the activity obtained in the 5 kb non-risk sequence, $n = 10$ independent experiments. **$p = 0.0014$, two-tailed paired Student's $t$-test. **d** In vivo zebrafish transgenic reporter assay. Green fluorescent protein (GFP) expression 24 h post fertilization (hpf) in mosaic F0 embryos injected with vectors containing a control sequence (top panel) or 5 kb interval sequence (bottom panel). **e** Comparison between 5 kb constructs containing risk or non-risk haplotype for enhancer-blocking property. Data are presented as the midbrain/somites EGFP intensity ratio of risk and non-risk sequences compared to empty gateway vector which has no enhancer-blocking activity ($n = 67$ for risk, $n = 73$ for non-risk, and $n = 87$ for control, biologically independent animals). Boxplot center line, median; box limits, upper and lower quartiles; whiskers, 1.5 × interquartile range and data beyond that threshold indicated as outliers. ****$p < 0.0001$; *$p = 0.049$, one-way ANOVA with pos hoc Holm-Sidak multiple correction test. Source data are provided as a Source Data file.

between fragments containing either the risk or non-risk haplotypes, with the risk fragment showing weakened enhancer-blocking activity (Fig. 3c).

To test this enhancer-blocking property in vivo we used a zebrafish reporter assay[30]. The 5 kb region was cloned in a reporter cassette containing green fluorescent protein (GFP) driven by a cardiac actin promoter and a midbrain enhancer. An enhancer-blocking element cloned in this vector would restrict the access of the midbrain enhancer to the GFP reporter gene, while the ability of the actin promoter to activate GFP in skeletal muscle and somites would be maintained (Fig. 3d). Unlike the episomal transfection in K562 cells, the zebrafish assay uses a transposon element to randomly integrate the test vector into the genome, which allows interrogation of sequence function in the genomic context and also analysis of a larger number of transgenic fish (50–150), each harboring independent genomic integrations of the tested construct.

In these experiments, the 5 kb region led to a decreased midbrain-specific GFP signal when compared to the control sequence, which displayed no enhancer-blocking activity (Fig. 3e; $p < 0.001$, one-way ANOVA with post hoc Holm–Sidak multiple correction test). Similar to the data obtained in the in vitro luciferase experiments, there was a significant difference in GFP activity between fragments containing the risk or non-risk alleles ($p = 0.049$, one-way ANOVA with post hoc Holm–Sidak multiple correction test), with the risk alleles resulting in reduced enhancer-blocking activity. Taken together, our data provide evidence that the 5 kb region has enhancer-blocking properties in vitro and in vivo, and that alleles of asthma-associated SNPs within this region are able to modulate this property.

**Genetic fine mapping and functional assays identify rs1888909 as likely causal variant**. We next attempted to identify the variant(s) likely mediating the allelic-specific differences we

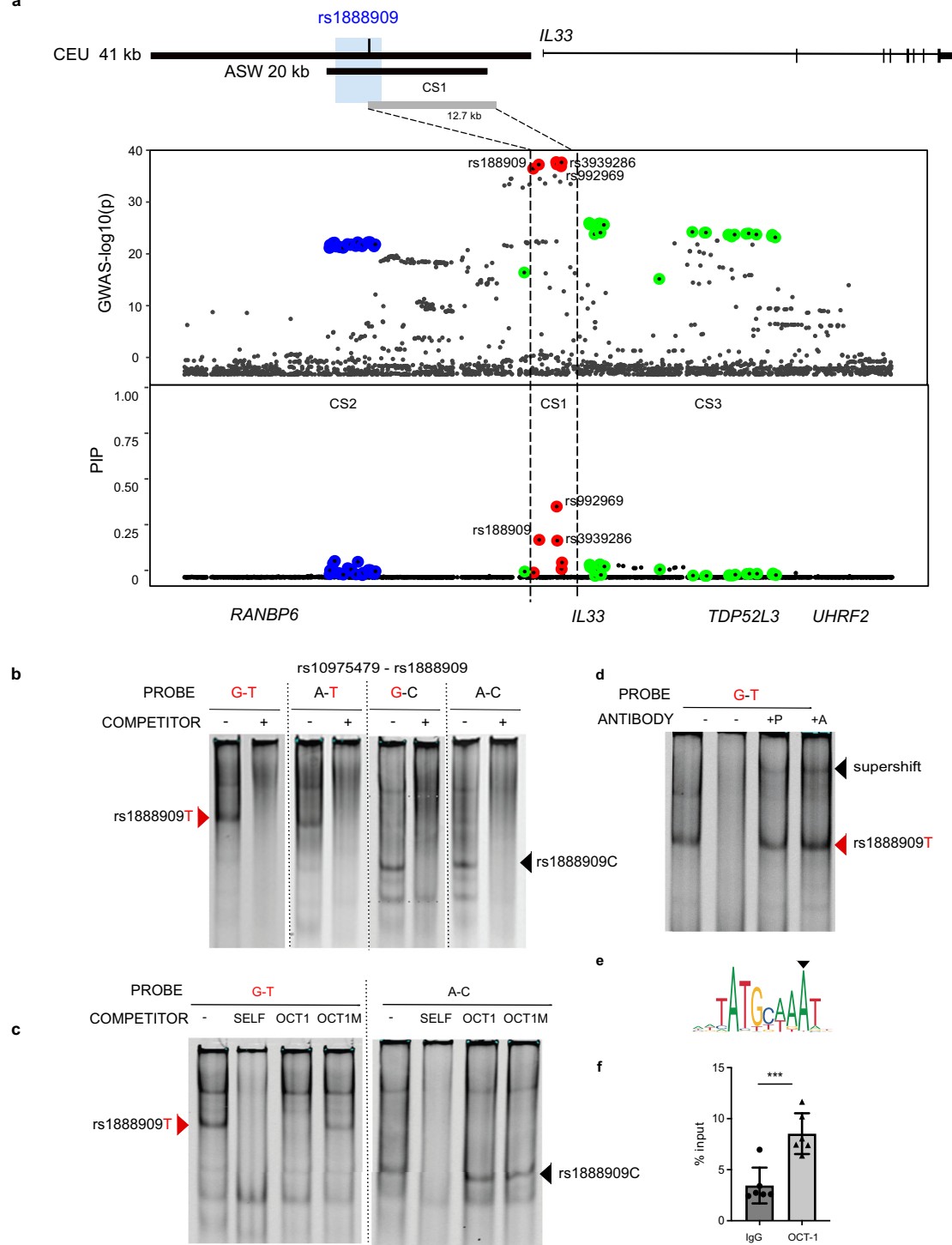

identified in our functional assays. We started by using a Bayesian analytical framework, sum of single effects (SuSiE)[31], to fine-map the associated region and identify credible sets (CS) of SNPs at the *IL33* locus with high probabilities of being causal. Within each CS, variants are assigned posterior inclusion probabilities (PIPs), with higher PIPs reflecting higher probabilities of being causal and the sum of all PIPs within a CS always equaling 1. Using all SNPs at this locus from a GWAS of childhood-onset asthma[2] in British white subjects from the UK Biobank[32], we identified three CSs (Fig. 4a and Supplementary Table 2). CS1 contained variants

with the highest PIPs and included six SNPs, all among those defined by LD with the lead GWAS SNPs, defining a 12.7 kb region (chr9: 6,197,392–6,210,099; hg19). The three CS1 SNPs with the highest PIPs included the lead SNPs in two multi-ancestry GWAS (rs992969; PIP 0.391), the lead SNP in the African-American GWAS[16] (rs1888909; PIP 0.209), and a SNP in LD with the lead SNPs (rs3939286; PIP 0.205). Only SNP rs1888909 overlap with the 5 kb region defined by functional genomic analysis and demonstrated to contain regulatory activity. We note that two other CSs of 25 SNPs (CS2) and 28 SNPs (CS3)

**Fig. 4 Potential causal GWAS SNP rs1888909T selectively binds OCT-1. a** Significance of SNP association in the GWAS[2] (top) and fine mapping results (bottom) for each variant at the *IL33* locus using SuSiE. Colors indicate each credible set (CS) identified. CS1: red, CS2 blue; CS3 green. CS1 (rs992969, rs1888909, rs3939286) defines a region of 12.7 kb (chr9: 6,197,392–6,210,099; hg19). Rs1888909 is the only variant that overlaps to the 5 kb region of interest (blue shade; chr9: 6,194,500–6,199,500; hg19). **b** Radiolabeled probes carrying the risk (in red) and/or non-risk sequences for SNPs rs10975479 and rs1888909 were incubated with nuclear extract obtained from K562 cells. Different complexes formed by rs1888909 are marked by red or black arrows. **c** Cold competition assay with OCT-1 consensus (OCT1) or mutated OCT-1 (OCT1M) oligonucleotides. EMSA probes and oligo competitor (100× molar excess) are noted above each gel. **d** Supershift complex formation with addition of anti-OCT-1 antibody as indicated by the black arrow. +P indicates addition of probe with nuclear extract followed by incubation with antibody. +A indicates incubation of extract with antibody followed by addition of probe;. **e** OCT-1 binding motif (JASPAR; MA0785.1) **f** Chromatin immunoprecipitation of H292 chromatin with anti-OCT-1 antibody followed by qPCR (ChIP-PCR). Plot shows enrichment of OCT-1 binding compared to input chromatin (IgG). Data are presented as mean ± SD ($n = 2$ independent experiments in triplicate each). ***$P = 0.0004$, one-tailed unpaired Student's *t*-test. Source data are provided as a Source Data file.

were identified by SuSiE, suggesting additional, independent regions potentially regulating the expression of *IL33* or other genes.

Focusing on the only SNP within the 5 kb region, we hypothesized that the risk allele of rs1888909 may alter transcription factor binding, resulting in altered regulatory activity and *IL33* expression. While rs10975479 was not predicted to be a causal variant, we also studied this SNP as it is only 15 bp away from rs1888909 (Fig. 1b). We performed an electrophoretic mobility shift assay (EMSA) using small labeled DNA probes and unlabeled competitors spanning four different combinations of the risk or non-risk alleles for variants rs10975479 (G/A) and rs1888909 (T/C). Upon incubation with nuclear extract, we observed differences in binding patterns between the risk (G–T) and non-risk (A–C) probes, suggesting that different transcription factor-binding complexes are formed in the presence of these two alleles (Fig. 4b). The major changes in binding are observed with both probes carrying the rs1888909 (T) allele, thereby indicating that the difference is driven by the risk allele T of this SNP. These data demonstrate that risk allele (T) of the asthma-associated SNP rs1888909 alters protein-binding properties, possibly influencing *IL33* expression.

To identify nuclear proteins that bind to the rs1888909 (T) vs rs1888909 (C) probes, we isolated three bands from the non-risk rs1888909 (C) lane and two bands from the risk rs1888909 (T) lane for mass spectrometry (MS) analysis (Supplementary Fig. 6). We filtered out the proteins that were also found in the control lane (cold probe competitor) and nuclear proteins that are not known to bind to DNA (Supplementary Table 3). We identified three transcription factors with known DNA-binding motifs bound only to the non-risk probe (NFE2, TFCP2, and FOXL2) and four transcription factors bound only to the risk probe (POU2F1, FOXP1, STAT3, and STAT5b). To select the transcription factors that bind to at least one octamer containing a risk or non-risk allele, we used the UniPROBE protein array database[33]. This analysis resulted in one transcription factor: OCT-1 (POU2F1), which bound to the EMSA G-T probe and was differentially bound to the risk allele. Using these orthogonal tools, we identified OCT-1 as our primary candidate for further investigation.

To confirm OCT-1-specific binding to the risk probe we first used a cold competition assay and demonstrated that the band specifically competed with an OCT-1 canonical DNA-binding motif, but not with a mutated oligonucleotide. Conversely, binding to the non-risk A–C probe was not competed by the consensus or mutated OCT-1 oligonucleotides, demonstrating specificity of OCT-1 association with only the risk allele (Fig. 4c). Further, an OCT-1-specific antibody supershifted the nuclear complex formed with the rs1888909 (T) probe (Fig. 4d). We were able to visualize this shift independently of the order of incubation of the nuclear extract with probe or antibody,

suggesting robust protein binding to the risk motif (Fig. 4e). We then performed chromatin immunoprecipitation (ChIP) followed by qPCR to demonstrate enrichment of OCT-1 binding to the DNA region containing the rs18889099 (T) (Fig. 4f).

We also investigated possible binding of several other candidate transcription factor obtained by MS or computationally predicted to bind to the EMSA probe containing the risk alleles (Supplementary Table 4 and Supplementary Fig. 7), We did not observe supershift when antibody was used against MAX, USF1, HIF1, DEC1, c-MYC, YY1, n-MYC, FOXP1, and STAT3.

These experiments provide strong evidence that OCT-1 binds differentially to the risk allele and directs the formation of differential allelic nuclear complexes at the rs1888909 variant.

**Asthma-associated SNPs are associated with *IL33* mRNA and IL-33 protein levels**. Finally, we tested the functional consequences of the GWAS SNPs rs1888909, rs10975479, and rs992969 on *IL33* mRNA and IL-33 protein abundance. The first two SNPs are within the 5 kb regulatory region. The latter, rs992969, is located outside of this region, but was previously reported to be associated with *IL33* expression in bronchial epithelial cells from primarily non-Hispanic white subjects[34]. SNP rs992969 is in high LD with rs1888909 in European ancestry populations ($r^2 = 1$ in CEU), but less so in African-American populations ($r^2 = 0.45$ in ASW), and in low LD with rs10975479 in both populations ($r^2 = 0.56$ and $0.13$, respectively) (Supplementary Fig. 1 and Supplementary Table 1). We used RNA-seq data from endobronchial brushings obtained from 124 asthmatic and non-asthmatic adult subjects, mostly of European ancestry (Fig. 5a, upper panel), and from nasal epithelial cell brushings from 189 African-American children from high-risk asthma families (Fig. 5a, lower panel). In both populations, carriers of one or two copies of the rs1888909 (T) risk allele had significantly higher *IL33* transcript levels compared to non-carriers of this allele. Genotypes at rs992969 were more modestly associated with transcript levels and only significant in the African-American samples after correcting for multiple (3) tests; while rs10975479 was not associated with *IL33* transcript abundance in either sample (Fig. 5a and Supplementary Table 5). These results are consistent with the EMSA data suggesting that allelic variants of rs1888909 result in differential protein binding. The stronger effects of rs1888909 on *IL33* expression in both populations suggest that rs1888909 is the causal variant in this region and that associations with other variants (such as rs992969) in GWAS and in gene expression studies are due to LD with the causal variant. We also determined that rs1888909 is an eQTL only for *IL33*, and not for neighboring genes, in nasal epithelial cells (Supplementary Table 6). This finding is in agreement with the genome organization data showing that *IL33* is the only expressed gene present on the same sub-TAD harboring the 5 kb region of interest containing rs1888909 (Supplementary Fig. 8).

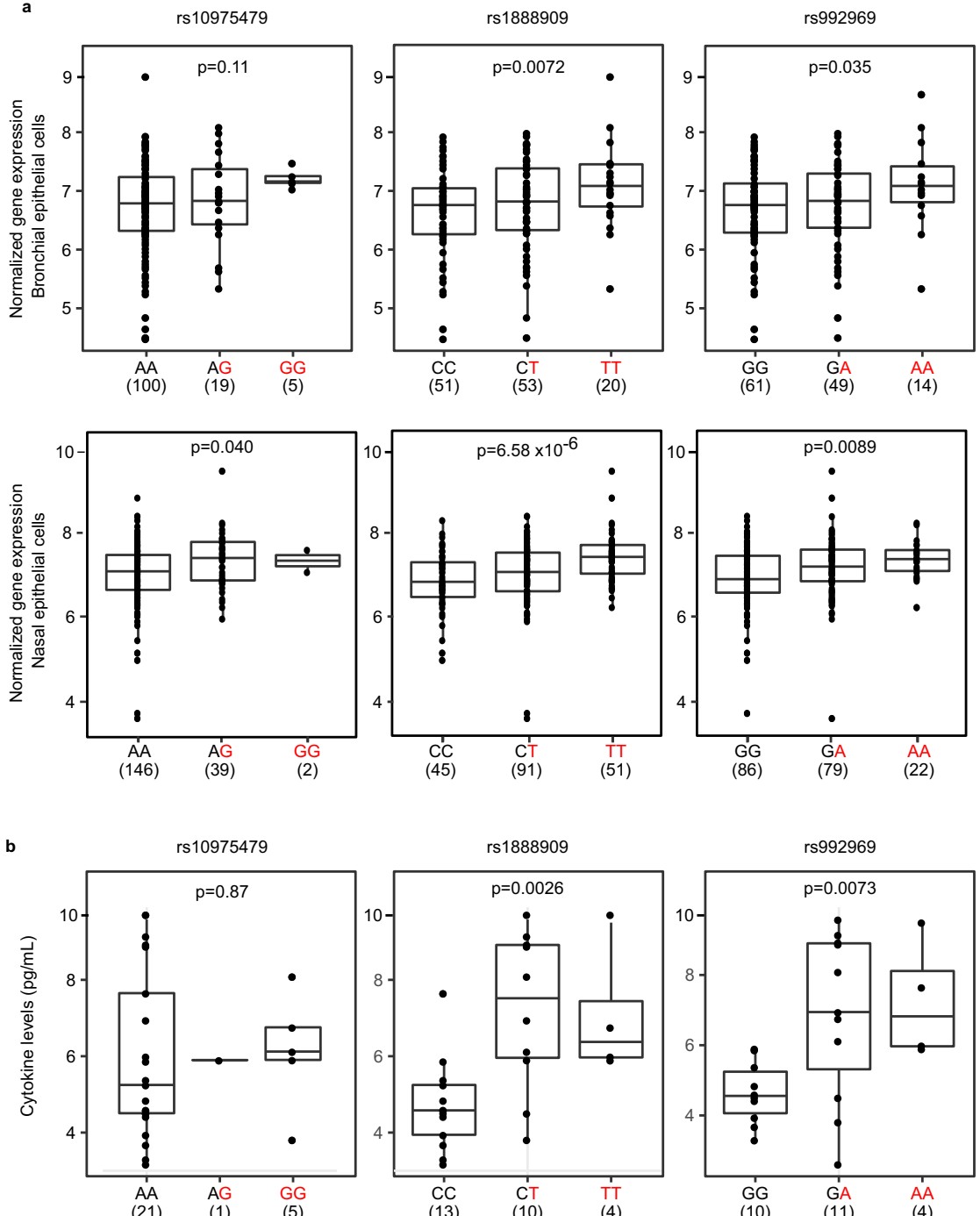

**Fig. 5 The rs1888909 (T) and rs992969 (A) alleles are associated with increased IL33 expression and IL-33 protein levels. a** Comparison of *IL33* expression between genotypes for SNPs rs10975479, rs1888909, and rs992969 from bronchial epithelial cells from 124 asthmatic and non-asthmatic adult subjects, mostly of European ancestry (upper panels) and nasal epithelial cells from 189 African-American children from high risk asthma families (lower panels). **b** Comparison of IL-33 cytokine levels between genotypes for SNPs rs10975479 ($n = 27$), rs1888909 ($n = 27$), and rs992969 ($n = 25$) measured in plasma from Hutterite children (all European ancestry). The asthma-associated risk allele at each SNP is highlighted in red (*x*-axis). The number of subjects per group is shown below the genotype. Boxplot center line: median; box limits: upper and lower quartiles; whiskers: 1.5× interquartile range. Statistical significance was determined using an additive linear model. Source data are provided as a Source Data file.

We observed similar patterns of association in studies of IL-33 protein levels in plasma from 30 children of European ancestry (Fig. 5b). Children who carried the asthma risk allele at rs1888909 or rs992969 had more IL-33 protein compared to children not carrying these alleles. There was no association between genotype at SNP rs10975479 and IL-33 protein levels.

The aggregate of our mouse transgenic assays, in vivo and in vitro reporter assays, and EMSA collectively support a function for rs1888909 as a potential causal variant for the association with *IL33* expression. The associations between the risk alleles and increased *IL33* transcript levels and IL-33 protein abundance further support a function for this region in regulating the *IL33*

gene and point to rs1888909 as a causal variant in this region, corroborating predictions from our in vitro and in vivo studies.

## Discussion

We described an integrated pipeline to fine-map and functionally annotate the asthma-associated locus that includes the *IL33* gene. We used the LD structure at this locus across populations of different ethnicities combined with a Bayesian fine-mapping tool to define a critical 20 kb genomic interval containing candidate causal SNPs for the asthma association. Epigenetic signatures further reduced this region to 5 kb, which we demonstrated to have a crucial function in the development of chromatin loops creating contacts between *IL33* promoters and regulatory elements within the critical interval to control *IL33* expression. Associations between rs1888909 (T) allele copies and increased *IL33* mRNA and IL-33 protein levels further suggests that this regulatory variant mediates the development of asthma among individuals carrying this risk allele.

A loss of function assay showed that this region is necessary for *IL33* expression in vivo, suggestive of an enhancer function. However, we failed to detect enhancer activity in reporter assays in cells where the region displays chromatin markers associated with enhancers. Our limited experimental conditions, including a small subset of cell types and reporter assays using a heterologous promoter, may explain why we failed to identify enhancer functions associated with this region. Moreover, the ENCODE project profiled the binding of hundreds of transcription factors across multiple cell types and reported no binding of transcription activators, transcriptional co-factors, RNA Pol II, or other factors usually associated with enhancers within this 5 kb region. Rather, the region is bound by CTCF and cohesin in most cell lines assayed by ENCODE. We showed that this region possesses enhancer-blocking activity in vivo and in vitro, reminiscent of insulator activities, and we were able to dissect its functional properties, to fine-map a variant that is likely causal to the association with asthma and to identify the molecular effector binding to this regulatory variant.

CTCF has a critical function in chromatin loop formation and participates in demarcation of Topological Association Domains (TADs). Loop formation within TADs facilitates contacts between specific genes and enhancers, allowing for appropriate temporal and tissue-specific expression[35]. We observed that the 5 kb risk haplotype carrying the asthma-associated allele possessed a significant loss of function, demonstrating a requirement for factors binding in that region for its proper regulatory activity.

We identified OCT-1 as a transcription factor that binds differentially to the risk allele of rs1888909, thus implicating OCT-1 in the function(s) of this 5 kb region. OCT-1 can regulate gene expression both positively and negatively. It has been shown to bind enhancers and regulatory regions upstream of multiple cytokines, including IL-3[36], IL-12p40[37], IL-13[38], IL-4[39], and IL-17[40] contributing to inducible chromatin remodeling. OCT-1 acts at hypersensitive sites by interacting with other families of proteins responsible for communicating with specific factors bound at the promoter or with the general transcription machinery. The mechanisms by which OCT-1 functions are strikingly diverse, and there are numerous studies reporting the coordinated activity with CTCF protein. For example, at the IL-17 cluster on chromosome 1, Oct-1 and CTCF facilitate long-range associations with the Th2 locus in naïve T cells. In a parallel fashion, it is possible that OCT-1 coordinates with CTCF to direct *IL33* expression at the locus through creation of appropriate chromatin interactions between the 5 kb asthma-associated region and the *IL33* promoter, which might explain the loss of E2-Crimson expression observed in h*IL33*^Crim^BAC5kdel mice (Fig. 2b–d). The

distinct binding complexes formed on the risk and non-risk alleles at the rs1888909 genomic region may be responsible for alterations in these interactions, which in turn may lead to altered *IL33* expression levels.

While our data support a function of rs1888909 in the regulation of *IL33* expression through OCT-1 binding and its primary association with asthma risk, our fine mapping results identified two other potential causative sets of SNPs associated with asthma at the *IL33* locus, upstream and downstream to the region that we studied (Fig. 1b). This suggests that other variants in this locus may have independent effects that alter the function of other regulatory elements and potentially control expression of *IL33* or other genes. Future studies will be needed to further dissect the mechanisms by which OCT-1 regulates *IL33* expression and to dissect the genetic architecture at this complex locus in order to address this hypothesis.

In summary, we identified a small 5 kb noncoding interval which is integral to *IL33* expression in several cell types. We propose a model in which CTCF mediates interaction of the 5 kb region to the *IL33* promoters through the formation of chromatin loops. The interplay between the distinct regulatory elements in the locus promotes spatial and temporal-specific regulation which is affected by OCT-1 binding to rs1888909. A deeper characterization of these mechanisms will not only enhance our understanding of *IL33* regulation but may also open new strategies in directing future studies, as well as other common diseases in which dysregulation of IL-33 is involved.

## Methods

**Experimental animals**. All CD-1 mice (Crl:CD1 IGS (ICR), strain code 022) were originally obtained from Charles River Laboratories, Inc. (Wilmington, MA, USA). Generation of h*IL33*^Crm^ BAC and h*IL33*^Crm^ 5Kdel transgenic mice was performed by the University of Chicago Transgenic Core Facility. Modified DNA was diluted to a concentration of 2 ng/μl and used for pronuclear injections of CD1 embryos. Procedures were conducted with approval of the Institutional Animal Care and Use Committee (IACUC) of University of Chicago (ACUP-71656; IBC0934) in accordance with standard protocols approved by the University of Chicago. All mice were housed with food and water ad libitum, temperatures of 65–75 °F (~18–23 °C) with 40–60% humidity, and a 12-h light/12-h dark cycle. Adult mice aged >8 weeks were used for reporter gene analysis.

**Red/ET BAC modification**. BAC RP11-725F15, obtained from the BACPAC Resource Center (Oakland, CA), was modified in vitro using the RED/ET recombination kit (Gene Bridges cat# ID: K002) according to the manufacturer's instructions. The reporter cassette was inserted in frame with the ATG of the human *IL33* gene, to replace the second exon of the gene, while maintaining fully intact boundaries and flanking regions. RED/ET plasmid was electroporated into BAC RP11-725F15 containing bacterial culture (chloramphenicol resistant) and plated for an overnight incubation on LB agar containing 3 μg/mL of tetracycline and 12.5 μg/mL of chloramphenicol at 30 °C. Subsequent colonies were grown into LB cultures containing the above concentration of antibiotics at 30 °C. A total of 30 μL of overnight culture was inoculated into 1.4 mL of LB culture containing the above concentration of antibiotics in a 2-mL microcentrifuge tube with the lid punctured. Cultures were grown at 30 °C until OD_600 ~0.3. Then 50 μL of 10% L-arabinose was added and the cultures were allowed to grow at 37 °C for 1 h. Primers, modified by 50 nucleotides of overhanging sequence with homology with the *IL33* start site, were designed to amplify a Crimson-kanamycin cassette from the pE2-Crimson-N1 vector. PCR reactions were purified using the QIAquick PCR Purification Kit (Qiagen) and dialyzed for 1 h using a Millipore MF-Membrane Filters (0.025 μm) in sterile water (Sigma). A total of 200 ng of purified PCR product was electroporated into BAC containing RED/ET plasmid cultures with L-arabinose and plated on LB agar containing 12.5 μg/mL of chloramphenicol and 20 μg/mL of kanamycin and grown overnight at 37 °C. Resulting colonies were analyzed for accurate recombination using restriction enzyme fingerprinting. Primers containing 50 bp BAC-homology arms used for generation of the recombination cassette are listed in Supplementary Table 7.

The asthma-associated 5 kb interval (chr9: 6,194,500–6,199,500; hg19) was deleted using RED/ET recombination kit (Gene Bridges, cat# ID: K002). The 5 kb region of interest was replaced by the ampicillin gene using primers containing 50pb homology arms flanking the region to be deleted. All correctly modified BACs were verified by fingerprinting and sequencing. BAC DNA was extracted using the Nucleobond PC20 kit (Macherey-Nagel; cat# 740571) and diluted for pronuclear injection. BAC copy number was determined as previously described[41].

**Tissue preparation, immunofluorescence staining, and microscopy**. Mouse lymph nodes or trachea were embedded into blocks with optimal cutting temperature compound (OCT 4583) and stored at −80 °C. Lymph nodes were not fixed prior to embedding, but were fixed on the slides just prior to permeabilization on the day of staining. Tracheas were fixed and bathed in a sucrose solution prior to embedding and freezing. Frozen tissues were sliced into sections 5–7-mm-thick and dried onto slides overnight. Sections were permeabilized, quenched, and blocked. Tissues were immunostained with the primary antibodies rat anti-mouse CD31: Biotin clone 390 for lymph nodes (1:250, Biolegend cat#102404) or Armenian hamster anti-mouse CD31 (1:500, tracheas, clone 2H8, Thermo Fisher Scientific, cat# MA3105), and rabbit anti-E2-Crimson (1:500, Living Colors DsRed Polyclonal Antibody, Takara Bio, cat# 632496). Sections were washed and stained with the secondary antibodies Streptavidin:Alexa Fluor 488 (1:250, Biolegend, cat# 405235) or goat anti-Armenian hamster IgG:AlexFluor 568 (1:1000, AbCam, cat# ab175716), and Goat anti-Rabbit IgG:Alexa Fluor 633 (1:1000, Thermo Fisher Scientific, cat# A-21070), along with the nucleic acid stain Hoechst 33342 (Thermo Fisher Scientific, cat# P36961). Coverslips were set with ProLong Diamond Antifade Mountant (Thermo Fisher Scientific, cat# 62249). Antibodies clones and dilutions used are listed in Supplementary Table 8.

Imaging was performed at the University of Chicago Integrated Light Microscopy Facility. Lymph node images were captured with a Leica TCS SP2 laser scanning confocal microscope (Leica Microsystems, Inc.) using a ×63/1.4 UV oil immersion objective and LAS_AF acquisition software (Leica Microsystems, Inc.). Trachea images were captured with a Leica SP8 laser scanning confocal microscope (Leica Microsystems, Inc.) using a ×20/0.7 multi-immersion objective and LAS_X Leica acquisition software. Further processing was completed using ImageJ software v1.53j (NIH, https://ij.imjoy.io/)

**Gene expression by real-time PCR**. Total RNA were isolated from control and insulator deleted mice using TRI-reagent (Sigma, cat# 93289). cDNA synthesis was performed using SuperScript II First-Strand Synthesis System (Thermo Fisher Scientific). Real-time qPCR reactions for E2-Crimson was performed using SsoAdvanced Universal SYBR Green Supermix (Bio-Rad Laboratories, Cat No. 1725270). Each PCR reaction contained 12.5 ng of reverse-transcribed RNA, 0.5 μM of each specific primer, SYBR Green PCR Master Mix, and RNase-free water to a final volume of 10 μL. Cycling parameters consisted of 2 min at 95 °C, followed by 40 cycles at 95 °C for 5 s and 58 °C for 30 s, and a melting step (55 to 95 °C, increment 0.5 °C for 5 s) was performed after each run to further confirm the specificity of the products and the absence of primer dimers. Relative gene expression was determined using cyclophilin A (peptidylprolyl isomerase A: *Ppia*; NM 008907) as an endogenous internal control. RNA samples with no reverse transcriptase added were used to test for genomic contamination. Primer sequence were: E2-Crimson (F: 5′-GCCAAG CTGCAAGTGACCAA-3′ and R: 5′-GCCTTGGAGCCGTAGAAGAA-3′) and Ppia (F: 5′-AATGCTGGACCAAACACAAA-3′ and R: 5′-CCTTCTTTCACCTTCCCA AA-3′).

**Construct preparation**. In all, 5 kb risk and non-risk PCR fragments were cloned into the pDONR vector (Thermo Fisher Scientific, cat# 12536017) and sequenced by the University of Chicago DNA Sequencing and Genotyping facility. DNA was subcloned into the enhancer-blocking vector[29] (gift from Dr. Laura Elnitski, NHGRI) or pGL4.23[luc2/minP] for enhancer assays (Promega, cat# E8411) using Gateway BP clonase (Thermo Fisher Scientific, cat# 11789013). Constructs were prepared using the Plasmid MidiPrep Kit (Qiagen, Cat. No./ID: 12143) and re-sequenced to confirm genotype. Constructs coordinates (hg19) are: 5 kb non-risk or risk; chr9: 6,194,675–6,199,500 and 5 kb control; chr7: 35,303,890–35,309,030.

**Luciferase assay**. K562 cells were transfected using TransIT-2020 Transfection Reagent (Mirus Bio, MIR 5404). Briefly, $10^5$ cells per well were plated 24 h prior to transfection. The transfection mix included 0.5 μg of enhancer blocking or PGL4.23 plasmid, 10 ng of pGL4.73 Renilla control vector, and 1.5 μl transfection reagent. TeloHAECs were transfected using JetPRIME transfection reagent (Polyplus, cat# 114-01). $2 \times 10^4$ cells per well in 24-well plates were transfected 3 days later with 250 ng of test plasmid DNA and 25 ng of pGL4.73 Renilla control vector (Promega, #E6911). After 48 h cells were harvested and assayed for luciferase activity using the Dual Luciferase Reporter Assay System (Promega, cat #E1910). Firefly luciferase was normalized to the pGL4.73[hRluc/SV40] Renilla luciferase (Promega, #E6911). A minimum of three independent transfections using different DNA preparations each round was assayed. A control plasmid containing the SV40 promoter and enhancer sequences driving luciferase expression was used as a positive control of transfection. Statistical significance of differences between groups was determined using GraphPad Prism (v 9.2.0).

**Zebrafish transgenesis**. The Tol2 vector contains a strong midbrain enhancer, a Gateway entry site, and the cardiac actin promoter controlling the expression of EGFP, and was developed to screen for insulator activity[30]. Each candidate sequence was recombined between the midbrain enhancer and the cardiac actin promoter. As a reference, the empty backbone was used (INS-zero). One cell-stage embryos were injected with 3–5 nL of a solution containing 25 nM of each construct plus 25 nM of Tol2 mRNA. Embryos where then incubated at 28 °C and

EGFP expression was evaluated 24 hpf. The midbrain/somites EGFP intensity ratio was quantified using ImageJ freeware (https://imagej.nih.gov/ij/) and was directly proportional to the enhancer-blocking capacity. As a positive control, the chicken beta-globin insulator 5HS4 was used. Each experiment was repeated independently and double-blinded to the operators.

**Genetic fine-mapping of the *IL33* locus**. We used SuSiE to perform genetic fine-mapping of the *IL33* locus for childhood-onset asthma in individuals from the UK Biobank (https://zenodo.org/record/3248979#.YTTfap5KjUJ). We used the same inclusion criteria and genotype QC measures as reported for Pividori et al. Childhood-onset asthma was defined as onset <12 yo (n = 9432) and controls as having no reported asthma at latest age of study (n = 318,167). We extracted genotypes from the *IL33* locus defined by Pividori et al. using the rbgen (v0.1) package in R (v3.6.1). Sex and the first 10 ancestral PCs were regressed out of the genotypes and phenotype. We assumed at most five causal variables when running susieR (v0.9.0).

**EMSA**. Nuclear extracts were prepared from K562 cells using NE-PER Nuclear and Cytoplasmic Extraction Kit (Thermo Fisher Scientific, cat# 78833) supplemented with HALT Protease Inhibitor Cocktail and PMSF (Thermo Fisher, cat# 78430). Protein concentrations were determined with the BCA kit (Thermo Fisher, cat# 23227). Oligonucleotides to be used as probes were synthesized with a 5′IRDye 700 modification (IDT). Binding reactions were performed using the Odyssey EMSA kit (LI-COR Biosciences, cat# 829-07310) and contained 5 μg nuclear extract, 2.5 nM labeled probe, and 100× excess unlabeled oligonucleotide (cold competitor) when noted. For supershift assays, 2 μg Oct-1 antibody (Santa Cruz Biotechnology, sc-232) was added and incubated for another 20 min. Reaction mixtures were run on a 4% nondenaturing polyacrylamide gel and analyzed with the LI-COR Odyssey Imaging System.

**Oligonucleotide/Probe sequences**. rs10975479G:rs1888909T: TCTGATGCAGAA CAGCAATGTGTTTTCCATGTGCACTTGGTC
  rs10975479G:rs1888909C: CCTGATGCAGAACAGCAATGTGTTTTCCACG TGCACTTGGTC
  rs10975479A:rs1888909T: TCTGATGCAGAACAACAATGTGTTTTCCATG TGCACTTGGTC
  rs10975479A:rs1888909C: CCTGATGCAGAACAACAATGTGTTTTCCACG TGCACTTGGTC
  Consensus OCT-1: TGTCGAATGCAAATCACTAGAA
  Mutant OCT-1: TGTCGAATGCAAGCCACTAGAA.
  The underlined sequences denote the precise location of the critical nucleotides being tested.

**Mass spectrophotometry and protein identification**. The sections to be analyzed from the EMSA gel were excised, washed, and destained using 100 mM $NH_4HCO_3$ in 50% aceto0nitrile (ACN). Sections then underwent reduction, alkylation, and trypsinization. Peptides were extracted with 5% formic acid, followed by 75% ACN in formic acid, and cleaned up with C-18 spin columns (Pierce). The samples were analyzed via electrospray tandem spectrometry (LC-MS/MS) on a Thermo Q-Exactive Orbitrap mass spectrometer at the Proteomics Core at Mayo Clinic, Rochester, Minnesota. Tandem mass spectra were extracted and then analyzed by Mascot and X!Tandem algorithms. Scaffold v4.8.4 (Proteome Software Inc.) was used to validate the protein identifications. Peptide identifications were accepted if they could be established at greater than 98% probability to achieve an FDR less than 1.0%.

**Candidate transcription factor analysis**. To computationally predict allelic specific binding of transcription factors to sequence containing variants of candidate SNPs we used an input sequence of 16 nucleotides centering on rs10975479 with GCAGAA-CA[A/G]CAATGTGT and rs18889098 with C or T to GTTTTCCA[C/T]GTGCA CTT. TRANSFAC database was used for the identification of putative transcription factor binding to motifs containing each variant (http://algten.lsi.upc.es).

After EMSA-mass spectrometry, Panther pathway analysis database[42] was used to filter for nucleic acid-binding proteins bound to the risk or non-risk probe. Jaspar (http://jaspar.genereg.net) and Alggen-promo (http://algten.lsi.upc.es) databases were used to screen these factors for known DNA-binding sites. Transcription factors that were also present on the control lane (cold competitor) were considered unspecific and excluded from the analysis.

To confirm differential binding to SNP rs1888909, we downloaded binding data for random octamers from UniProbe (http://thebrain.bwh.harvard.edu/uniprobe/downloads/All/All_Contig8mers.zip) and identified all transcription factors that bound at least one octamer containing a risk or non-risk allele.

**Chromatin immunoprecipitation**. ChIP assays were performed using ChIP assay kit (Millipore, cat# 17-925) and protocol. $2 \times 10^6$ H292 cells were fixed with 1% formaldehyde. Chromatin was sonicated using a diagenode BioRuptor. Lysates were incubated overnight with 5 μg of Oct-1 (Santa Cruz, sc-232x) or an IgG control (Santa Cruz, sc-2027). Following the washes, elution of chromatin

complexes, and reversal of crosslinks, DNA was recovered using QIAquick pcr purification kit (Qiagen, Cat. No./ID: 28104). Input DNA was simultaneously processed. Enrichment for OCT-1 binding was determined by qPCR using SsoAdvanced Universal SYBR Green Supermix (Bio-Rad Laboratories, Cat No. 1725270) and primers: F: 5′-GCCTCTGGTCTCAGTGGATA-3′ and R: 5′-CTGCTCATAGGAGACACAGTAAAG-3′.

**Gene expression and genotype studies**. RNA-seq data from airway epithelial cells were available from two sources. The first was from bronchial epithelial cells sampled from adults (75 European American, 45 African American, 4 other), with and without asthma (83 cases, 41 controls), who participated in a study of asthma in Chicago (Chicago Asthma Genetics Study/ CAG). Procedures for bronchoscopy, cell and RNA processing, and genotyping have been previously described[26]. For this study, normalized gene expression counts were adjusted for age, sex, current smoking status, sequencing pool, the first three ancestry PCs.

The second source was from nasal epithelial cells sampled from 187 African-American children (71 cases, 98 controls,18 unknown) from a birth cohort of children at high risk for asthma (Urban Environment and Childhood Asthma/ URECA)[43]. Procedures for nasal brushing and cell processing followed standard procedures[44]. Genotypes were determined using Illumina Multi-Ethnic Genotyping Array (MEGA), and processed using standard QC[26]. To test for association between genotypes for three SNPs and *IL33* transcript levels, we used an additive effects linear model, including as covariates sex, study site, batch id, epithelial cell proportion, and 12 latent factors in the epithelial cell studies. Latent factors were included to correct for unwanted variation[45].

For both datasets, linear regression considering additive genotype effects on gene expression was performed using limma in R (v3.3.3). *P* values were adjusted for three tests; *P* < 0.016 was considered significant. All subjects provided written informed consent and children provided written assent; these studies were approved by University Chicago Institutional Review Board.

**IL-33 cytokine and genotype studies**. Blood for cytokine studies was drawn from 28 Hutterite children during trips to South Dakota, as previously described[46]. Written consent was obtained from the parents and written assent was obtained from the children. The study was approved by the institutional review boards at the University of Chicago.

Briefly, The Milliplex Map Human TH17 Magnetic Bead Panel (Millipore, cat# MTH17MAG) was used to measure IL-33 levels in thawed supernatants (plasma) at the University of Chicago Immunology Core facility using standard protocols, as previously described[47]. Associations between genotypes at rs1888909, rs992969, and rs10975479 and IL-33 cytokine abundance were tested using a linear model, assuming additive effects. Genotypes were obtained using PRIMAL50, an in-house pedigree-based imputation tool that imputes variants from whole-genome sequences from individuals who were genotyped with an Affymetrix genotyping array. One subject had poor IL-33 cytokine data and was removed. Of the remaining subjects, 27 subjects for rs10975479 and rs1888909 and 25 subjects for rs992969 had good-quality genotype calls and were retained. Linear regression between the genotypes and IL-33 cytokine levels was performed using limma, adjusting for sex and age.

**Reporting summary**. Further information on research design is available in the Nature Research Reporting Summary linked to this article.

## Data availability
All data supporting the findings of this study are available within the article and Supplementary Information file or from the corresponding author upon reasonable request. Source data are provided with this paper. Publicly available datasets used in this manuscript: UniProbe—http://thebrain.bwh.harvard.edu/uniprobe/downloads/All/All_Contig8mers.zip All_Contig8mers.zip file provided in the link above contain enrichment scores of transcription factor binding to random nucleotide k-mers obtained in protein biding microarray (PBM) experiments. Relevant files for each protein in the database are under "Downloads". UK Biobank—https://zenodo.org/record/3248979#.YTTfap5KjUJ. This is a dataset from the UK Biobank of individuals with and without asthma, using data for genome-wide and transcriptome-wide studies, identifying both shared and distinct genetic risk factors for childhood and adult onset asthma. Promoter Capture Hi-C sequences used in this study were from the GEO database under primary accession codes GSE152550 (AEC) and GSE79718 (LCL). Source data are provided with this paper.

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

## Acknowledgements
We thank Don Wolfgeher for assistance with sample preparation and analysis of the mass spectrophotometry; Dr. Laura Elnitski for generously supplying the enhancer barrier plasmids; Dr. Benjamin Glick for supplying the E2-Crimson reporter vector; Dr. Anna Di Rienzo for the TeloHAECs and Dr. Francois Spitz for comments and valuable suggestions. This work was supported by NIH grants R01 HL118758, R01 HL128075, R01 HL119577, R01 HL085197, U19 AI095230, UG3 OD023282 and UM1 AI114271.

## Author contributions
M.A.N., A.I.S., D.C.D. and I.A conceived and supervised the study. I.A. and D.C.D designed and performed experiments. D.R.S. and L.E.M performed the Hi-C experiments and analysis. N.J.S. performed candidate transcription factor analysis. K.M.B performed mouse immunofluorescence staining and microscopy; C.L. Hrusch performed luciferase assays, C.L. Howard performed mouse BAC transgenic experiments. J.T. and J.L.G.-S performed and analyzed the zebrafish experiment. C.O and N.S. contributed reagent. K.M.M. and S.M.C. performed fine-mapping RNA and protein QTL analysis. M.M.S processed sample for cytokine studies. J.E.G., D.J.J., M.C.A., E.T.N., D.K.H. and S.R.W. provided patient samples. I.A., D.C.D., M.A.N. and A.I.S. wrote the manuscript with comments from all the authors.

## Competing interests
The authors declare no competing interests.
