## [Peer Review File · Nature Communications]

Asthma-associated genetic variants induce IL33 differential expression through an enhancer blocking regulatory regionREVIEWER COMMENTS

Reviewer #1 (Remarks to the Author):

Aneas et al have identified a region in the IL-33 locus that acts as a regulatory element both in vitro and in vivo. They show that variants of IL-33 that have been associated with asthma mediate allele-specific regulatory activity as well as IL-33 expression.

1. In Fig 2 why look at staining specifically in the Lymph Node? There is clear distinction in endothelial cells and agree this exhibits difference in mouse and human nicely – but why not also show expression in the lung? It should be expressed in basal cells in the lung. Does expression change if you induce allergic inflammation in the mice – or with other stimuli that have been shown to elicit the release of IL-33 in vivo?

2. From Fig 4 the conclusion is that there is regulatory region upstream of the IL33 locus and implicate the asthma-associated SNPs rs10975479 and rs1888909 in regulating IL33 expression. The argument would be strengthened if the authors could examine the functional consequences of this regulation

3. In Fig 6 was there any correlation with asthma occurrence/severity in these cohorts in 6a and b?

4. Several statements in discussion overplay the results;

Line 501-506 – very speculative regarding the function of OCT-1

Line 520: “results together with the increased IL33 expression observed in humans with the risk allele offer a plausible mechanistic explanation for the association of these variants and asthma risk.” I don't agree that the authors have provided mechanism.

Reviewer #2 (Remarks to the Author):

The article ‘Asthma-associated genetic variants induce IL33 differential expression through a novel regulatory region’ dissects the asthma associated loci near the gene encoding for the IL33 cytokine. This region is of relevant to asthma as models indicate elevated IL33 levels. The authors utilize numerous datasets to pinpoint a 5kb locus, and more particularly a key SNP within that locus, that is in a possible regulatory element governing IL33 expression to be associated with asthma. The authors went on and characterize the regulatory nature of the 5kbp locus over IL33 expression in vivo using a transgenic mice model. They conclude that the nature of regulation asserted is not as classical enhancer, nor insulator, but rather an intra-TAD enhancer-promoter divider/blocker. The authors went deeper and characterized the interaction between a candidate TF, OCT-1, the 5kb locus genetic sequence and the IL33 expression phenotype, to find that the asthma-associated SNP reduces the interaction of OCT-1 and release of enhancer-promoter blockage. Overall, the paper is well written, coherent and easy to follow. I particularly appreciate the use of several independent population genetic datasets to logically reduce the genetic search field to start with. I believe the paper is a perfect fit for the broad audience of Nature Communications following the following revisions.

Major points

-For BAC experiments, both wild-type and deletion, would be good to get number of mouse lines tested for each one in text and how variable number of integration or site of integration can be on expression.

-How many CTCF binding peaks are there in ENCODE in the 5kb region? In how many cell lines are they observed (i.e. finding CTCF peaks in more cell lines is usually more indicative of a functional CTCF site)? What is their orientation compared to IL33? More info on this is needed in text.

-In the discussion the authors called the 5kb locus ‘defies the standard definition of regulatory elements’ promoting the idea of localizing enhancer blocking regulatory entity. The authors should

elaborate further why this element is not considered by them a regular insulator.

-More work needs to be done to rule out distance between promoter and enhancer versus insulator function as having an effect on studies. Figure 4a. The desert construct, although it does not contain an insulator, is illustrated as shorter than the 5kb region. The drop in expression could be explained in part by the increase distance. While no p value is provided for the CONTROL-DESERT expression comparison on the looks of it seems significantly different. The decrease in construct expression could decline in a non-linear manner with linear increase in distance. Same for Figure 4e – It will be beneficial to include the full size 5kb fragment reporter construct in this comparison. How can the authors distinguish, as mention before, between differential effects of getting the enhancer closer to the promoter, rather than the true regulatory nature of the inserted fragment. If fragment size were maintained, please clarify this.

Minor points

- 'acts as a strong regulatory element' might be good to mention what type of regulatory element here. I realize definition is problematic, but even a more general one could be helpful.

- 'We show that genotype' not clear what you mean here. Maybe 'We show that the asthma-associated SNP, rs1888909....

- 'Heterochromatin is a highly compacted region in the genome and not actively involved in gene regulation' the fact that this region is silenced could be involved in regulating a gene, i.e. silencing an enhancer or other. Would not completely rule this out and provide that caveat in text.

-Would be good to analyze Hi-C, Hi-ChIP and PLAC-seq datasets from human to check whether these regions interact. I realize these will be different cell types, but can further suggest/support that there is an interaction here.

-The authors choose to focus on the role of OCT-1 binding through the 5kb regulatory region and the effect of the indicated SNPs on its function/regulation of IL33. OCT-1 was shown in the past to be involved in asthma through regulation of other genes as well. As deeper discussion on the operation of OCT-1 with other asthma related genes and more specifically the effect of associated SNPs on those interaction might be helpful. This might shed more light on the unique regulatory nature suggested here.

-Figure 1 – the authors should consider a clearer illustration (UCSC style tracks) of the LD block and the follow of zoom-in-zoom-out along sections a->c

-Figure 1c – please add names / codes of cell name in the Chromatin state track

-Figure 3a – add a small legend in the figure for the epigenome roadmap cell types.

-Line 278 – change "RNA PolI" to "RNA Pol II"

-Line 315 – add a reference to Fig.4e

-Line 370-372 – If possible, please explain why adding this additional filtration step. One could see the shortlisting of TF (not found in the control lane, non-DNA-binding and differential between risk and non-risk allele) as enough. For example, both FOXP1 and STAT3 could be interesting candidates to elaborate on as they are implicated in lung epithelial development and refractory asthma.

-Figure 5d – Once OCT-1 binding had become the main focus, I find ChIP-qPCR or ChIP-seq even much more convincing than competition assay. In Fig.5d it would be beneficial to show enrichment of OCT-1 binding between the risk and non-risked allele. Also the background for this is yellow versus the others being white. Would change to white.

-Fig.6a – add an indication of the origins of the upper and lower panel samples.

Reviewer #3 (Remarks to the Author):

In this work the Authors investigated genetic risk factors linked to asthma. Starting from SNPs that clusters in individuals diagnosed with Asthma, they choose a lead candidate based on literature and GWAS data (IL-33 locus).

Further, they narrow the locus, crossing databases from different ethnicity to obtain multiples SNPs clustered in 3 CSs. Following leads, they reduce the common region to a 5Kb sequences found upstream of the IL33 gene. Next assessing their findings, they used a humanized Mice model transduced with BAC constructs including the full gene and regulatory regions +/- the 5kb seq. This assay revealed the requirement of the 5kb region for appropriate expression of the gene

in that model. However, exploration of the chromatin structure by 4C felt short without a clear-cut picture. Nevertheless, the authors perused their work in the understanding of the 5Kb region using classical reporter assays and then a construct made for in vivo investigation in zebrafish. From these, they reduced the 5Kb region to single SNPs ultimately leading to a 400bp containing two rs that were relevant in the GWAS databases.

At last, the author worked toward the functionality of the sequence. Leading hypothesis being that (i) the 5kb region loops to allow expression of the IL-33 gene, (ii) the 400bp within the 5Kb is central and can be summarized by two rs and act as a blocker (iii) in individual with Asthma, something else might bind through the motifs, resulting in an overall altered expression of IL 33. Using EMSA and Mass spec they identified OCT-1, suggesting an increase of IL-33 expression due to the enrichment of this transcription factor. Finally, the author use a wealth of RNA Seq data to correlates the SNPs to expression of IL-33 and confirm their hypothesis in vivo for one rs (e.g 1888909).

Altogether this lengthy study jump from one model to another (in order :mice, K562, H292 cell line, zebrafish and transcriptomic databases) and work around multiple rs (but focusing on three) giving the average reader a tedious overview and a challenging time to grasp all the data presented and in which systems and variety of terms used GWAS, SNPs, LD and specially rs to a point that even the Authors are making typos (See l. 386).

Hence my remarks are both on the comprehension of this work, that could use additional tables to summarized the findings and biological resources used, and on some technical overview and aspects of the study.

Line 75 the Authors mention that linkage study found a region 2.3kb upstream IL 33. later they mention SNPs at 72.3Kb up the gene and later 22kb from the TSS of the same gene. I am curious if any additional thoughts were brought towards the first region at 2.3Kb (there is no mention after this part). Is there any overlap between this region and the CSs presented? What is the score associated with that SNPs (using SuSIE)?

L131 to 141, hypothesis is made towards the chromatin landscape according to its epigenetic features available in the literature. Did the Authors explored their region using the 3D genome browser, based from Rao et al., 2014 : <http://promoter.bx.psu.edu/hi-c/tutorial.html> ? i.e are they LADs and subTADS at play in that region. If so that will simplify the writing of this work, as rs modulating borders of subtads.

In order to get closer to physiological, condition the author used a humanized mice, transduced with BACS, the mention of four independent lines L 202, refers at four lines from four different mice?

Regarding the 4C experiment, an interesting choice of technique considering the model, it is difficult to make a conclusion based on the data presented:

In the text, the bait is located upstream the 5kb del. A smart choice to encompass any modification of chromatin structure at this region. However looking at the figure presented, very few things are going on: only one blue interaction disappear with the deletion, and strangely one can observe two interactions within the deletion..It is well known that 4C techniques produce huge background and noise in sequencing. Here, the figure presented seems to reflect this effect: overall identical shape, but with a 'lower' level in the deletion assay.

Without knowing to what the assay was normalized one cannot simply conclude.

Since the author use BACs constructs, digestion, ligation in vitro of the sole construct represent one of the minimal controls. In the same lines of observations, how many replicates were used, how many sequencing performed, average depth and number of reads retrieved? Even if trendy, 4C assays are cumbersome and prone to multiple bias (see Raviram et al., 2014 for example), without validation by 3C (using different digestion enzymes)/ 3D-FISH /confronting data to the available database (Rao et al.,) this assay feels like an overshoot. Conclusion made by the authors here, are not supported by their data (no statistical test provided, only one blue arc disappeared in the BACs -5Kb construct)

Can the author also detail their choice/rationale for dendritic cells? (why not using fresh tissues).

Next, investigations are done regarding the 5kb seq in reporter assays. One can read the results

as enhancer blocking activity, not due to the rs but other seq found in the 5kb regions; CTCF boarders for example. The control used without this seq sounds unperfect (does not seem to be the same size as well). A better control would be the WT region (as used) and for example a random piece of seq in between the CTCF sites.

In Fig4a and b, it is not precise if the different construct reported are the same, performed multiple times, different ones, each using one risk rs; one using all risk associated rs or even a combination of both. Such assay highly depends on environmental factors including but not limited to cells confluency, number of insertions, place of insertions,... Details need to be provided to appreciate the (expected) variability presented in 4b.

Next experiment uses an in vivo approach in zebrafish. Here again the rationale behind the model (zebra) is missing. Why not using a yeast system? More efficient and less time consuming for a question that is limited to expression under the control of epigenetic phenomenon. In the same line, a cell system can be used, allowing one to stay in a cellular cell type closest to the physiological one. Here the system is performed in zebrafishes, using the property of developmental tissues (somites/ midbrain). The authors need to better argument what seems like a waste of biological resources and money at this point.

Besides, the controls seem to be incomplete (1 CTCF, cut sequences, disparities in length of sequences inserted) and not well justified. Why only use a control with 1 CTCF site, when the authors know CTCF sites "work in pairs"; doing so they might lose crucial insights. Here, the constructs reported used a combination of two rs, in their "risk associated version". How does this correlates with human data? Does these two rs segregate in Individuals diagnosed with Asthma. Or is this a way to push the reporter system one way, to deliver a clean readout? (fine, but need to be justified).

On the EMSA, author nicely work around what can bind to the rs sequences, combining with MS to identify the probable actors involved. Prior to the MS, I wonder if investigations on the rs were made using traditional in silico approaches: how the rs (even if limited to a sub section of the one detailed earlier), behave when using the HOCOMOCO Human v11 (<https://hocomoco11.autosome.ru/>). Any overlap with the TF found later in MS? Interestingly, further validation of OCT1 is performed using ChIP on H292 cells, what is the haplotype of rs associated in these cells?

Looking at the last figure presented, one wonders how it compares when using all RSs described in the first part of the study. The confirmation and delineation of mechanism around the rs1888909 is elegant. Finding molecular partners linked to this rs, and illustrating the effect using RNA Seq on patients is surely powerful and straight forward. In fact, this assessment should be performed using all rs, including the 2.3kb upstream IL 33 and the three SNPs mentioned at beginning (L148) to reinforce the observations made by the authors. If the rs1888909 is a key player, the other rs should not present significant up regulation of the IL 33 mark. At least not as much.

Similarly, in this figure data is presented regarding the rs992969. This rs present a significant correlation with IL 33 levels, no EMSA was performed using this sequence, why? The authors have a straight forward setup to asses if this rs can act with OCT-1. Even if naïve, this hypothesis can be rapidly controlled.

On the other hand data presented for the rs10975479, suggest this rs to be irrelevant in the pathway described by the author. Thus, the driver, according to the results presented only relies on rs1888909 (what is the proportion of this rs in the general pop and in the individuals diagnosed with Asthma in their different database?). Looking back at data from Fig4, it really feels like the lack of constructs is a mis-opportunity to consolidate this statement.

If overall interesting, the cumbersome means deployed (unsurprisingly, the 4C assay is not informative) hurt the fluidity of the work.

Without changing their message, the data could be better presented, some assay moved to supplemental (e.g. the 4C), rationale of models chosen explicitly detailed and confrontation to human transcriptomics highlighted (Fig6). Including a higher evaluation of all rs mentioned in their first paragraph, hypothetically under the control of other factors/ mechanisms that the rs1888909.

This study gather an impressive wealth of data, ultimately focusing on one rs. Nevertheless, improvement can and should be made on the delivery: details in experiments are missing, some complementary assay and analysis should be performed to strengthen the main claim.

Reviewer #4 (Remarks to the Author):

This paper describes a series of experimental approaches which have attempted to identify the causal variant(s) driving the GWAS signal at the IL33 locus for asthma. The authors initially defined credible variant sets, used bio-informatic approaches to try and narrow down the key regulatory region variants, and then used a range of approaches including expression in transgenic mice and zebrafish models, cell line and ex vivo asthma airway studies to study the region of interest.

1 The paper focuses mainly on rs1888909. This variant appears to be most important in the African American population. Using Open Targets to interrogate this variant interestingly the evidence mapping this variant to IL33 is weak: eQTL and pQTL analyses point to other genes in the region, and the enhancer signal again doesn't point at IL33. ERMP1 has the strongest eQTL evidence for this SNP. This does not diminish the work contained in this paper but it does imply that using current variant to gene mapping approaches would not lead to high confidence this variant is potentially producing an effect through altered IL33 expression. This is important because much of the experimental work in the paper is based on the assumption that it is altered IL33 expression which is underlying the effect seen. The only good evidence to support this comes from the ex vivo studies of 124 individuals. The analyses presented are however pooled for the analyses shown in figure 6, with the populations containing both asthma and non-asthma individuals or being from pooled ethnic groups. The pQTL analysis in particular is small; it should be easy to recruit additional subjects to enlarge this. I assume the IL33 assay used has been carefully evaluated for serum samples: measuring IL33 in serum is well known to be a difficult assay to use.

2 Given the above issue, the main results deal with the potential regulatory region containing the rs1888909 variant. The authors provide convincing data from both transgenic mouse and zebrafish models that this region of the genome contains regulatory activity, although given it has OCT-1 binding activity this isn't that surprising, and of course does not prove that the region is actively involved in gene regulation in human cells when present in the context of the surrounding genome. A more usual way to approach this would now be to use CRISP/Cas9 to manipulate the genome in situ and then study the consequences by for example RNAseq in a relevant cell type (eg primary human bronchial epithelial cells). However given the time it must have taken to create all the transgenic models and characterise these I can see that the work may to some extent pre-date the wider use of CRISP/Cas9 approaches. To be fair, the work performed is extensive and original. The weakest part to my mind is the TeloHAEC cell model, in which the haplotypes were expressed in a luc reporter but unfortunately no enhancer/repressor effects were seen unless a rather artificial system was used to try and increase activity using SV40/HS2.

3 The study utilises a wide variety of tissue types, which rather confuses the overall story. If the effect of rs1888909 is really being driven by altered IL33 expression in airway epithelium, why use transgenic models where the read outs are endothelium or in the mid brain? Obviously zebrafish don't have true lungs but an epithelial read out would seem to be most relevant for a gene product which is an alarmin and given the airway epithelial cell data presented.

4 Given the eQTL or other data (Hi-C, TSS/enhancer mapping) suggesting ERMP1 or UHRF2 are probable targets for this variant (see Open Targets), it would be useful to add analyses looking at these genes to the ex vivo datasets.

Response to reviewers' comments

We would like to thank the 4 reviewers for their extensive suggestions about our work. Following these recommendations, we reworked our manuscript to streamline it, render the storyline more linear and strengthen our conclusions by new experiments, adding new data that further support our conclusions. We believe that our manuscript is much stronger and easier to read thanks to the reviewers' suggestions. In the end, our conclusions remain the same, but is further bolstered by our new data. Below we present a point-by-point response to each comment,

Reviewer #1 (Remarks to the Author):

Aneas et al have identified a region in the IL-33 locus that acts as a regulatory element both in vitro and in vivo. They show that variants of IL-33 that have been associated with asthma mediate allele-specific regulatory activity as well as IL-33 expression.

1. In Fig 2 why look at staining specifically in the Lymph Node? There is clear distinction in endothelial cells and agree this exhibits difference in mouse and human nicely – but why not also show expression in the lung? It should be expressed in basal cells in the lung. Does expression change if you induce allergic inflammation in the mice – or with other stimuli that have been shown to elicit the release of IL-33 in vivo?

Thank you for these comments. First, we wanted to show that our “humanized” mice containing the full BAC (*hIL33^{Crim}* BAC) had a human tissue specific expression pattern of the reporter gene. In humans, the original cloning and description of IL33 expression was in the high endothelial venules (HEV), which are where cells can enter the lymph nodes directly from the blood. Interestingly, mouse IL-33 is not produced in the HEV tissue. However, the reviewer is correct that we should also see expression and protein production of human IL-33 in the basal cells of the lungs. Unlike humans who have basal epithelial cells throughout the lungs, mouse basal cells are primarily limited to the epithelium in the trachea. Thus, to address the reviewer's inquiry, we performed new experiments using trachea of the transgenic mice stained for the reporter. As shown in the revised Figure 2b and 2c below, mouse basal epithelial cells from the *hIL33^{Crim}* BAC transgenic mice express the Crimson reporter specifically in the basal epithelium of the trachea. As in the lymph node, the *hIL33^{Crim}* BAC5kdel mice fail to express the reporter. (lines 151-168, page 7)

We are in the process of preparing a separate manuscript on the effects of both LPS treatment and allergic inflammation on lung expression of the human IL-33 reporter. Because of the extensive amounts of additional data that we have for that manuscript, we felt that adding it to this manuscript would make it unfocused. We hope the reviewers agree that the genomics studies presented in this manuscript provides a coherent and important study on its own

2. From Fig 4 the conclusion is that there is regulatory region upstream of the IL33 locus and implicate the asthma-associated SNPs rs10975479 and rs1888909 in regulating IL33 expression. The argument would be strengthened if the authors could examine the functional consequences of this regulation

We agree with the reviewer. The functional follow-up experiments to unlock the molecular mechanisms by which *IL33* misregulation lead to asthma are ongoing. These include extensive, long-term in vivo and in vitro experiments, involving generating new mouse models, designing genome editing in appropriate human cell lines. Our goal for this study was to show an integrated pipeline that led to the identification of causal variants at the *IL33* locus. We feel that the aggregate data from reporter, EMSA and QTL studies (mRNA and protein) in 3 human cohorts collectively support our claims that rs1888909 is causally associated with asthma susceptibility.

3. In Fig 6 was there any correlation with asthma occurrence/severity in these cohorts in 6a and b?

This is an interesting question. Due to the small sample sizes of the cohorts, we were not properly powered to analyze the associations stratified by asthma status. Because these variants are robustly associated with asthma in nearly every GWAS we did not think it was necessary to further replicate that association in our study. Figure 6 was revised as figure 5.

**4. Several statements in discussion overplay the results;
Line 501-506 – very speculative regarding the function of OCT-1**

We rewrote this section to indicate that we do not know how OCT1 binding to this region might be mediating regulatory effects on IL33 and the information below was added to the discussion session (line 360-362, page 17).

“Future studies will be needed to further dissect the mechanisms by which OCT-1 regulates IL33 expression and to dissect the genetic architecture at this complex locus in order to address this hypothesis”

5. Line 520: “results together with the increased IL33 expression observed in humans with the risk allele offer a plausible mechanistic explanation for the association of these variants and asthma risk.” I don’t agree that the authors have provided mechanism.

We agree and reworded the text accordingly, toning down our conclusions and interpretation of our data.

Reviewer #2 (Remarks to the Author)

The article ‘Asthma-associated genetic variants induce IL33 differential expression through a novel regulatory region’ dissects the asthma associated loci near the gene encoding for the IL33 cytokine. This region is of relevant to asthma as models indicate elevated IL33 levels. The authors utilize numerous datasets to pinpoint a 5kb locus, and more particularly a key SNP within that locus, that is in a possible regulatory element governing IL33 expression to be associated with asthma. The authors went on and characterize the regulatory nature of the 5kbp locus over IL33 expression in vivo using a transgenic mice model. They conclude that the nature of regulation asserted is not as classical enhancer, nor insulator, but rather an intra-TAD enhancer-promoter divider/blocker. The authors went deeper and characterized the interaction between a candidate TF, OCT-1, the 5kb locus genetic sequence and the IL33 expression phenotype, to find that the asthma-associated SNP reduces the interaction of OCT-1 and release of enhancer-promoter blockage. Overall, the paper is well written, coherent and easy to follow. I particularly appreciate the use of several independent population genetic datasets to logically reduce the genetic search field to start with. I believe the paper is a perfect fit for the broad audience of Nature Communications following the following revisions.

Major points

6. For BAC experiments, both wild-type and deletion, would be good to get number of mouse lines tested for each one in text and how variable number of integration or site of integration can be on expression.

We appreciate the suggestion and the information below was added to manuscript (line 148-150 page 7)

“Estimated BAC copy number varied from 2-11 copies between founders. Copy number and Crimson expression for each line containing either the full BAC (n=5) or the 5kb deletion (n=4) is shown on Supplementary Figure 4”

7. How many CTCF binding peaks are there in ENCODE in the 5kb region? In how many cell lines are they observed (i.e. finding CTCF peaks in more cell lines is usually more indicative of a functional CTCF site)? What is their orientation compared to IL33? More info on this is needed in text.

There are 2 CTCF sites in the 5kb region (both in forward orientation) and the information below regarding how many cell lines contain each CTCF site was added to manuscript (line 110-112, page 5)

“We also identified two CTCF sites within 2kb of each other with evidence of CTCF binding in multiple cell lines (84 cell lines have CTCF binding to site 1 and 142 cell lines have CTCF binding to site 2, out 194 ENCODE-3 lines”.

We also added the orientation of all the CTCF sites in one megabase around this region to the new supplementary figure 8 (see response to Reviewer 3, question 24).

8. In the discussion the authors called the 5kb locus ‘defies the standard definition of regulatory elements’ promoting the idea of localizing enhancer blocking regulatory entity. The authors should elaborate further why this is element is not considered by them a regular insulator.

Taking into consideration all the concerns raised by the reviewers regarding the 4C experiment (see response to Reviewer 3- question 26, for more details), we decided to remove those data from the manuscript, which also resulted in the removal of an entire section. We worked on generating more appropriate constructs in order to properly test for the function of the 5kb region and strength the hypothesis of this sequence being an enhancer blocking element (see response to question 9). The discussion was modified and the statement raised by the reviewer was removed.

9. More work needs to be done to rule out distance between promoter and enhancer versus insulator function as having an effect on studies. Figure 4a. The desert construct, although it does not contain an insulator, it is illustrated as shorted than the 5kbp region. The drop in expression could be explained in part by the increase distance. While no p value is provided for the CONTROL-DESERT expression comparison on the looks of it seems significantly different. The decrease in construct expression could decline in a non-linear manner with linear increase in distance. Same for Figure 4e – It will be beneficial to include the full size 5kb fragment reporter construct in this comparison. How can the authors distinguish, as mention before, between differential effects of getting the enhancer closer to the promoter, rather than the true regulatory nature of the inserted fragment. If fragment size were maintained, please clarify this.

We appreciate the suggestion. In order to account for the distance between the enhancer and the promoter and its influence on luciferase activity, we created a new set of constructs. This included a control construct in which we cloned a 5kb fragment lacking any epigenetic marks or TF binding according to the ENCODE data (chr7:35303890-35309030). Constructs and haplotypes are shown in figure 3a (Figure 4 was revised as figure 3). For the first comparison between control and the 5kb region of interest, we prepared a total of 6 DNAs, 3 with 5kb controls and 3 with the 5kb region of interest from the *IL33* locus. We performed 3 different transfections in K562 cells using a control and 5kb region pair each time. All DNAs were transfected in triplicate well per plate. Using these rigorously paired and controlled experiments, we observe a 40% reduction in luciferase activity from the *IL33* construct when compared to the control DNA (Figure 3b, below). Panel c compares activity between the 5kb risk and non-risk haplotypes. For this, we used 10 different preparations of DNA for each haplotype. These results corroborate our previous findings but are better founded, and we thank the reviewer for raising this important issue.

Minor points

10. ‘acts as a strong regulatory element’ might be good to mention what type of regulatory element here. I realize definition is problematic, but even a more general one could be helpful.

Thank you for the suggestion and the information below was added to manuscript (*Line 45-46, page2*)

“Here, we identify a 5kb region within the GWAS-defined segment that acts as an enhancer barrier element in vivo and in vitro”.

11. We show that genotype’ not clear what you mean here. Maybe ‘We show that the asthma-associated SNP, rs1888909....

Thank you for the suggestion, we modified the text accordingly (*Line 48-51, page2*)

“We show that the asthma-associated SNP rs1888909, located within the 5kb region, is associated with IL33 gene expression in human airway epithelial cells and IL-33 protein expression in human plasma, potentially through differential binding of OCT-1 (POU2F1) to the asthma-risk allele”

12. Would be good to analyze Hi-C, Hi-ChIP and PLAC-seq datasets from human to check whether these regions interact. I realize these will be different cell types, but can further suggest/support that there is an interaction here.

We added the analysis suggested by both reviewers, and data from 4 different human Hi-C sources (LCL, Airway epithelial cells, HUVECs and K562) corroborate our original description

that the 5kb region loops to the promoter of *IL33* (line 124-131, page 6). These data are shown in Supplementary Figure 3. Also see comment 26 from reviewer 3.

13. The authors choose to focus on the role of OCT-1 binding through the 5kb regulatory region and the effect of the indicated SNPs on its function/regulation of IL33. OCT-1 was shown in the past to be involved in asthma through regulation of other genes as well. As deeper discussion on the operation of OCT-1 with other asthma related genes and more specifically the effect of associated SNPs on those interactions might be helpful. This might shed more light on the unique regulatory nature suggested here.

We appreciate the suggestion and more specifics were added to the discussion (Line 358-360, P17)

14. Figure 1 – the authors should consider a clearer illustration (UCSC style tracks) of the LD block and the follow of zoom-in-zoom-out along sections a->c

Genomic coordinates were added to the top panel of Figure 1.

15. Figure 1c – please add names / codes of cell name in the Chromatin state track

Cell names were added to this figure (now 1b).

16. Figure 3a – add a small legend in the figure for the epigenome roadmap cell types.

That figure was removed from the paper and replaced with HiC data in human cells from our lab and public databases (see Supplementary Figure 3).

17. Line 278 – change “RNA Poll” to “RNA Pol II”

We changed it in the manuscript (line 183, page 9 and line 327, page 15).

18. Line 315 – add a reference to Fig.4e

Panel for 4e was removed from the manuscript. Figure 4 was revised as figure 3.

19. Line 370-372 – If possible, please explain why adding this additional filtration step. One could see the shortlisting of TF (not found in the control lane, non-DNA-binding and differential between risk and non-risk allele) as enough. For example, both FOXP1 and

STAT3 could be interesting candidates to elaborate on as they are implicated in lung epithelial development and refractory asthma.

Uniprobe was used as an orthogonal experiment to validate of our MS findings. Even though, FOXP1 and STAT3 are interesting candidate bound to the risk probe, they are not predicted to bind directly to risk allele (see also answer to question 32 -Reviewer 3). Additional super shift EMSA gels for FOXP1 and several factors, including STAT3 were added as a new figure to supplementary material and were shown not to be directly bound to the probe (Supplementary Figure 7). Results below were added to the manuscript (lines 266-270, p12-13)

“We also investigated possible binding of several other candidate transcription factor obtained by MS or computationally predicted to bind to the EMSA probe containing the risk alleles (Supplementary Table 4; Supplementary Fig. 7), We didn't observe supershift when antibody was used against MAX, USF1, HIF1, DEC1, c-MYC, YY1, n-MYC, FOXP1 and STAT3”

20. Figure 5d – Once OCT-1 binding had become the main focus, I find ChIP-qPCR or ChIP-seq even much more convincing than competition assay. In Fig.5d it would be beneficial to show enrichment of OCT-1 binding between the risk and non-risked allele. Also the background for this is yellow versus the others being white. Would change to white.

We changed the background to white. Figure 5 was revised as Figure 4 (see panel 4f).

21. Fig.6a – add an indication of the origins of the upper and lower panel samples.

Thanks for the suggestion. Sample types were depicted on each panel. Figure 6 was revised as Figure 5.

Reviewer #3 (Remarks to the Author):

In this work the Authors investigated genetic risk factors linked to asthma. Starting from SNPs that clusters in individuals diagnosed with Asthma, they choose a lead candidate based on literature and GWAS data (IL-33 locus).

Further, they narrow the locus, crossing databases from different ethnicity to obtain multiples SNPs clustered in 3 CSs. Following leads, they reduce the common region to a 5Kb sequences found upstream of the IL33 gene. Next assessing their findings, they used a humanized Mice model transduced with BAC constructs including the full gene and regulatory regions +/- the 5kb seq. This assay revealed the requirement of the 5kb region for appropriate expression of the gene in that model. However, exploration of the chromatin structure by 4C felt short without a clear-cut picture. Nevertheless, the authors perused their work in the understanding of the 5Kb region using classical reporter assays and then a construct made for in vivo investigation in zebrafish. From these, they reduced the 5Kb region to single SNPs ultimately leading to a 400bp containing two rs that were relevant in the GWAS databases.

At last, the author worked toward the functionality of the sequence. Leading hypothesis being that (i) the 5kb region loops to allow expression of the IL-33 gene, (ii) the 400bp within the 5Kb is central and can be summarized by two rs and act as a blocker (iii) in individual with Asthma, something else might bind through the motifs, resulting in an overall altered expression of IL 33. Using EMSA and Mass spec they identified OCT-1, suggesting an increase of IL-33 expression due to the enrichment of this transcription factor. Finally, the author use a wealth of RNA Seq data to correlates the SNPs to expression of IL-33 and confirm their hypothesis in vivo for one rs (e.g 1888909).

Altogether this lengthy study jump from one model to another (in order :mice, K562, H292 cell line, zebrafish and transcriptomic databases) and work around multiple rs (but focusing on three) giving the average reader a tedious overview and a challenging time to grasp all the data presented and in which systems and variety of terms used GWAS, SNPs, LD and specially rs to a point that even the Authors are making typos (See l. 386).

22. Hence my remarks are both on the comprehension of this work, that could use additional tables to summarized the findings and biological resources used, and on some technical overview and aspects of the study.

We appreciate these comments from the reviewer. We reworked the story line extensively, making it more streamlined and linear. This entailed removing sub-sections (such as a section describing the 4C-seq in humanized mice), reorganizing the text and figures to display a more cohesive description of the results. We also generated more data that was added to the supplementary material. Details of each of these steps are discussed and illustrated below.

23. Line 75 the Authors mention that linkage study found a region 2.3kb upstream IL 33. later they mention SNPs at 72.3Kb up the gene and later 22kb from the TSS of the same gene.I am curious if any additional thoughts were brought towards the first region at 2.3Kb (there is no mention after this part). Is there any overlap between this region and the CSs presented?What is the score associated with that SNPs (using SuSIE)?

The asthma-associated SNPs from GWAS span a 40Kb DNA segment and the segment is located 2.3kb upstream of the gene (See Supplementary figure 1 for a list of all SNPs in high LD with at least one of the lead SNPs and their distance from the gene). There is no association between SNPs in this 2.3kb region with the disease. We agree that this was confusing, and revised the text, simplifying it. As a result, we do not mention the 2.3kb distance and reworked figure 1 such that each genomic coordinate or distances are clearly illustrated. We also decided to move the fine-mapping experiments to later in the manuscript, as they are directly relevant to zooming in on rs1888909, and this was somewhat lost in the previous version in which fine-mapping (using SuSiE) was in the opening of the results section.

24. L131 to 141, hypothesis is made towards the chromatin landscape according to its epigenetic features available in the literature. Did the Authors explored their region using the 3D genome browser, based from Rao et al., 2014 : <http://promoter.bx.psu.edu/hic/tutorial.html> ? i.e are they LADs and subTADS at play in that region. If so that will simplify the writing of this work, as rs modulating borders of subtads.

We thank the reviewer for the suggestion. We looked at the HiC data from Rao et al and used the 3D genome browser to show the TAD structure at the locus. As we illustrate in the new Supplementary Figure 8 (see below), SNP rs1888909 and the 2 CTCF sites (both in forward orientation) are located in the middle of a sub-TAD (highlighted in gray) and not on the borders. This figure also illustrates how this TAD contains only 2 genes, *IL33* and *TPD52L3*, with *IL33* being the only one expressed in nasal epithelial tissue and participating in long-range chromatin interactions with the 5kb region that is the focus of our study (see answer #26 and the new Supplementary Figure 3 below). These data and figures clearly strengthen our claim that the 5kb region is physically (and functionally, given in vivo the results of the deletion in the human BAC in humanized mice) connected to *IL33* expression.

25. In order to get closer to physiological, condition the author used a humanized mice, transduced with BACS, the mention of four independent lines L 202, refers at four lines from four different mice?

This is correct. Each transgenic line harbors a random genomic insertion of the BAC DNA, resulting that each stable transgenic line represents an independent biological replicate. The strong concordance between the various lines we analyzed strengthen the notion that the data we obtain is a result of the endogenous interactions within the transgenic BAC, rather than positional effects or other variables unique to a given transgenic line.

26. Regarding the 4C experiment, an interesting choice of technique considering the model, it is difficult to make a conclusion based on the data presented: In the text, the bait is located upstream the 5kb del. A smart choice to encompass any modification of chromatin structure at this region. However looking at the figure presented, very few things are going on: only one blue interaction disappear with the

deletion, and strangely one can observe two interactions within the deletion. It is well known that 4C techniques produce huge background and noise in sequencing. Here, the figure presented seems to reflect this effect: overall identical shape, but with a ‘lower’ level in the deletion assay. Without knowing to what the assay was normalized one cannot simply conclude. Since the author use BACs constructs, digestion, ligation in vitro of the sole construct represent one of the minimal controls. In the same lines of observations, how many replicates were used, how many sequencing performed, average depth and number of reads retrieved? Even if trendy, 4C assays are cumbersome and prone to multiple bias (see Raviram et al., 2014 for example), without validation by 3C (using different digestion enzymes)/ 3D-FISH /confronting data to the available database (Rao et al.,) this assay feels like an overshoot. Conclusion made by the authors here, are not supported by their data (no statistical test provided, only one blue arc disappeared in the BACs -5Kb construct). Can the author also detail their choice/rationale for dendritic cells? (why not using fresh tissues).

We agree with the reviewer. We followed the reviewer’s suggestion and generated corroborating evidence for the interaction between the 5kb region and *IL33* in 4 different human cell lines, including two from our lab (bronchial epithelial cells and LCLs) and two from public datasets suggested by the reviewer (HUVEC and K562, same cells that we use for functional assays in our manuscript). As such, we opted to remove the mouse 4C-seq data and the corresponding section from the manuscript for the sake of streamlining our narrative, adding several independent lines of evidence that the 5kb region interacts with *IL33* in human cell lines, forming the new Supplementary Figure 3 (displayed below). These new data show the same as our mouse data, but can be explained and illustrated in a much simpler way than the humanized mouse data. And, as the reviewer pointed, we did not have appropriate controls and sufficient replicates in the mouse to make the claims we did in the original submission.

27. Next, investigations are done regarding the 5kb seq in reporter assays. One can read the results as enhancer blocking activity, not due to the rs but other seq found in the 5kb regions; CTCF borders for example. The control used without this seq sounds unperfect (does not seem to be the same size as well). A better control would be the WT region (as used) and for example a random piece of seq in between the CTCF sites.

We thank the reviewer for bringing up this issue, which was also highlighted by reviewer 2. Please refer to our answer to question 9 (reviewer 2), above. We redesigned our constructs and retested them accordingly, with our new data still supporting our original claims.

28. In Fig4a and b, it is not precise if the different construct reported are the same, performed multiple times, different ones, each using one risk rs; one using all risk associated rs or even a combination of both. Such assay highly depends on environmental factors including but not limited to cells confluency, number of insertions, place of insertions,... Details need to be provided to appreciate the (expected) variability presented in 4b.

Figure 4 has now been revised as Figure 3. We thank the reviewer for their comment. We modified this figure and expanded on our narrative of what was precisely done in these experiments. We added a new panel to show the constructs used and the risk and non-risk haplotypes. In Figure 3b we use the non-risk haplotype for the 5kb region compared to the same

length control fragment. We performed 3 transfections using 6 different DNAs of the control and non-risk constructs. In Figure 3c we used the same non-risk and compared to the risk haplotype. We performed 10 independent transfections using 20 different DNA preparations of the non-risk and the risk constructs. Each construct was consistently transfected into 100,000 K562 cells in triplicate wells each round. These enhancer blocking constructs use the PGL3-basic vector backbone (Promega) which results in transient, non-integrating episomes when transfected into cells, which can explain the variability observed. This information has been now added to the figure legend (Figure 3, p34) and corresponding methods sub-section (line 455, p21).

29. Next experiment uses an in vivo approach in zebrafish. Here again the rationale behind the model (zebra) is missing. Why not using a yeast system? More efficient and less time consuming for a question that is limited to expression under the control of epigenetic phenomenon. In the same line, a cell system can be used, allowing one to stay in a cellular cell type closest to the physiological one. Here the system is performed in zebrafishes, using the property of developmental tissues (somites/ midbrain). The authors need to better argument what seems like a waste of biological resources and money at this point.

We appreciate the reviewer's suggestion. Unlike the episomal transfection in K562 cells, the zebrafish assay uses a transposon element, the *Tol2* system, to randomly integrate the test vector into the genome of the cell. This allows interrogation of sequence function in the genomic context and also analysis of a larger number of animals per construct injected. In fact, using a single round of injections in zebrafish early embryos we obtain 50-150 transgenic fish, each representing an independent stable transgenic integration. All embryos are housed in a single 90mm petri dish in an incubator in the lab for 24-48 hours before visualization and quantification of reporter expression. Thus, there is no more work involved than in a routine reporter assay in cells, with the advantage of an in vivo interrogation under chromatinized conditions. This information was added to the manuscript (line 203-206, p10).

30. Besides, the controls seem to be incomplete (1 CTCF, cut sequences, disparities in length of sequences inserted) and not well justified. Why only use a control with 1 CTCF site, when the authors know CTCF sites "work in pairs"; doing so they might lose crucial insights.

We agree with the reviewer. Because the 5kb constructs already point to regulatory properties and allele-specific effects of the haplotypes tested, we opted to simplify the figure, keeping only the data for the 5kb fragment. Further delineation of the functional impact of SNPs in this region were carried out in the later sections of the manuscript (genetic fine-mapping, differential binding of transcription factors, EMSA, eQTL/pQTL analyses, all highlighting rs1888909 as the likely causal variant of the association). This also resulted in a more streamlined presentation of our results.

31. Here, the constructs reported used a combination of two rs, in their “risk associated version”. How does this correlates with human data? Does these two rs segregate in Individuals diagnosed with Asthma. Or is this a way to push the reporter system one way, to deliver a clean readout? (fine, but need to be justified).

SNP rs10975479 has a much lower risk allele frequency in both European and African ancestry (0.15 and 0.08, respectively) compared to SNP 1888909 (0.52 and 0.76, respectively). Our 5kb risk construct contains the most common risk haplotype (rs10975479A: rs1888909T); the results are plotted in the new Figure 3a-b (previously numbered as figure 4). We also added allele frequencies for each candidate SNP to Supplementary Table 1 (the information for rs10975479 and rs1888909 displayed below).

chr	position	LD (r ²)	LD (D')	Variant	Ref	Alt	AFR freq	AMR freq	ASN freq	EUR fre
9	6197377	0.56	-1	rs10975479	A	G	0.08	0.13	0.00	0.15
9	6197392	0.98	0.99	rs1888909	T	C	0.52	0.78	0.97	0.76

32. On the EMSA, author nicely work around what can bind to the rs sequences, combining with MS to identify the probable actors involved. Prior to the MS, I wonder if investigations on the rs were made using traditional in silico approaches: how the rs (even if limited to a sub section of the one detailed earlier), behave when using the HOCOMOCO Human v11 (<https://hocomoco11.autosome.ru/>). Any overlap with the TF found later in MS?

We thank the reviewer for the suggestion. We looked for transcription factors (TF) predicted to bind sequences containing the risk and non-risk alleles for rs1888909 and rs10975479 using Transfac database (<http://algggen.lsi.upc.es>). The results were added to supplementary material. The only TF that overlapped with MS was OCT-1.

We also ran the tool <https://molotool.autosome.ru> on sequences containing the risk and non-risk alleles for rs1888909 and rs10975479. The tool requires selection of transcription factor binding sites (TFBS) to be searched, so we selected TFs from Supplementary Table 3 found in the MS experiment (POU2F1, FOXP1, STAT3, STAT5A, NFE2). Of these TFs, the tool does not have TFBS motifs for FoxL2 and TFCP2. Using a p-value cut-off of 1e-3, we identified binding of FOXP1 and POU2F1 to the probe, but only POU2F1 (OCT-1) was differentially bound to the risk allele. This differential binding was also confirmed using a third orthogonal tool, UNIPROBE (Universal PBM Resource for Oligonucleotide Binding Evaluation), a database which hosts data generated by universal protein binding microarray (PBM) technology on the in vitro DNA-binding specificities of proteins. These data are summarized on the table below.

Risk probe

GAACAACAATGTGTTTTCCA	TGTGCACTTGGTCAACACCTA	
TGTGTTTTTC		MoLoTool p = 2.559e-4 FOXP1
TGTTTTCCA	TGTGCAC	MoLoTool p = 3.945e-4 POU2F1
T.CA	TGCA	Uniprobe score=0.40669 Pou2F1

Non-risk probe

GAACAACAATGTGTTTTCCAC	GTGCACTTGGTCAACACCTA	
TGTGTTTTTC		MoLoTool p = 2.559e-4 FOXP1

Finally, we experimentally confirmed the differential binding of OCT-1 to these alleles by performing EMSA supershift assay for OCT-1, FOXP1 and several other factors that could putatively bind to the probe sequence predicted by Transfac. These new experiments are now described in the results and illustrated as a new Supplementary figure 7. Altogether, multiple orthogonal lines of evidence support the claim that allelic variants of rs1888909 result in differential OCT-1 binding, complementing the genetic fine-mapping (revised Figure 4a) and eQTL/pQTL data (Figure 5- previously numbered as figure 6) which point to support the same conclusion.

33. Interestingly, further validation of OCT1 is performed using ChIP on H292 cells, what is the haplotype of rs associated in these cells?

H292 cells are heterozygote for rs1888909.

34. Looking at the last figure presented, one wonders how it compares when using all RSs described in the first part of the study. The confirmation and delineation of mechanism around the rs1888909 is elegant. Finding molecular partners linked to this rs, and illustrating the effect using RNA Seq on patients is surely powerful and straight forward. In fact, this assessment should be performed using all rs, including the 2.3kb upstream IL 33 and the three SNPs mentioned at beginning (L148) to reinforce the observations made by the authors. If the rs1888909 is a key player, the other rs should not present significant up regulation of the IL 33 mark. At least not as much. Similarly, in this figure data is presented regarding the rs992969. This rs present a significant correlation with IL 33 levels, no EMSA was performed using this sequence, why? The authors have a straight forward setup to asses if this rs can act with OCT-1. Even if naïve, this hypothesis can be rapidly controlled.

The strategy we used to prioritize our candidate SNPs was first based on the local LD structure. Using LD to fine-map a complex trait is based on the premise that ancestral meiotic recombinations diminish LD. Next, we integrated functional genomic annotation using publicly available data to identify active chromatin regions, implying that the SNP with the strongest association with a trait might not be the causal variant, but be located close to the causal variant. SNPs **rs1342326** and **rs2381416** did not overlap with the 20kb LD-defined region based on African American population. SNP **rs992969** lies into a region of heterochromatin in virtually every Epigenome Roadmap cell lines (see figure below, which is the new Supplementary Figure 2, with the blue marker on each row indicating heterochromatin in the corresponding cell line). The parsimonious inference is that this SNP is not accessible for binding of transcription factors and unlikely to be a functionally causal SNP. SNPs rs1888909 and rs992969 are in very high LD ($r^2=0.98$), which might explain the positive correlation of both SNPs with *IL33* expression. However, only rs1888909 fulfills the criteria of being both in accessible chromatin and an eQTL for *IL33*. SNP rs10975479 was also included in the analysis for being only 15pb away from our candidate SNP, but as explained above and in the manuscript, has very low minor allele frequency, is not in LD with rs1888909 and does not have strong evidence as an eQTL for *IL33*.

35. On the other hand data presented for the rs10975479, suggest this rs to be irrelevant in the pathway described by the author. Thus, the driver, according to the results presented only relies on rs1888909 (what is the proportion of this rs in the general pop and in the individuals diagnosed with Asthma in their different database?). Looking back at data from Fig4, it really feels like the lack of constructs is a mis-opportunity to consolidate this statement.

Please refer to response to question # 31, above.

36. If overall interesting, the cumbersome means deployed (unsurprisingly, the 4C assay is not informative) hurt the fluidity of the work. Without changing their message, the data could be better presented, some assay moved to

supplemental (e.g. the 4C), rationale of models chosen explicitly detailed and confrontation to human transcriptomics highlighted (Fig6). Including a higher evaluation of all rs mentioned in their first paragraph, hypothetically under the control of other factors/mechanisms that the rs1888909.

We agree with the reviewer and thank them for their valuable comments, suggestions and criticisms, all of which we strived to address. We believe that the current version of the manuscript is significantly streamlined and has a more fluid narrative, while all the new experiments and analyses further supported our previous claims. We hope the reviewer agree with our assessment.

Reviewer #4 (Remarks to the Author):

This paper describes a series of experimental approaches which have attempted to identify the causal variant(s) driving the GWAS signal at the IL33 locus for asthma. The authors initially defined credible variant sets, used bio-informatic approaches to try and narrow down the key regulatory region variants, and then used a range of approaches including expression in transgenic mice and zebrafish models, cell line and ex vivo asthma airway studies to study the region of interest.

37. The paper focuses mainly on rs1888909. This variant appears to be most important in the African American population. Using Open Targets to interrogate this variant interestingly the evidence mapping this variant to IL33 is weak: eQTL and pQTL analyses point to other genes in the region, and the enhancer signal again doesn't point at IL33. ERMP1 has the strongest eQTL evidence for this SNP. This does not diminish the work contained in this paper but it does imply that using current variant to gene mapping approaches would not lead to high confidence this variant is potentially producing an effect through altered IL33 expression. This is important because much of the experimental work in the paper is based on the assumption that it is altered IL33 expression which is underlying the effect seen. The only good evidence to support this comes from the ex vivo studies of 124 individuals. The analyses presented are however pooled for the analyses shown in figure 6, with the populations containing both asthma and non-asthma individuals or being from pooled ethnic groups. The pQTL analysis in particular is small; it should be easy to recruit additional subjects to enlarge this. I assume the IL33 assay used has been carefully evaluated for serum samples: measuring IL33 in serum is well known to be a difficult assay to use.

Open target has eQTL evidence in blood only. There were no association in lung or any other relevant cell/tissue. To check this, we tested association between rs1888909 and other genes at the locus in nasal epithelial cells and show that SNP rs1888909 was only an eQTL for the *IL33* gene. (Please see data at our response to #40 below).

We added a statement in the discussion section acknowledging the possibility that rs1888909 and other functional variants in this locus may be impacting the expression of other genes, which can be the focus of future experiments (lines 358 – 366, p17).

“This suggests that other variants in this locus may have independent effects that alter the function of other regulatory elements and potentially control expression of *IL33* or other genes. Future studies will be needed to further dissect the mechanisms by which OCT-1 regulates *IL33* expression and to dissect the genetic architecture at this complex locus in order to address this hypothesis”

IL-33 cytokine assay was performed in plasma samples (line 565, p260).

38. Given the above issue, the main results deal with the potential regulatory region containing the rs1888909 variant. The authors provide convincing data from both transgenic mouse and zebrafish models that this region of the genome contains regulatory

activity, although given it has OCT-1 binding activity this isn't that surprising, and of course does not prove that the region is actively involved in gene regulation in human cells when present in the context of the surrounding genome. A more usual way to approach this would now be to use CRISP/Cas9 to manipulate the genome in situ and then study the consequences by for example RNAseq in a relevant cell type (eg primary human bronchial epithelial cells). However given the time it must have taken to create all the transgenic models and characterize these I can see that the work may to some extent pre-date the wider use of CRISP/Cas9 approaches. To be fair, the work performed is extensive and original. The weakest part to my mind is the TeloHAEC cell model, in which the haplotypes were expressed in a luc reporter but unfortunately no enhancer/repressor effects were seen unless a rather artificial system was used to try and increase activity using SV40/HS2.

We appreciate your comment and we acknowledge that the use an artificial system might be a limitation to the enhancer assay. In our study, 5kb fragments containing the risk and non-risk haplotypes were cloned upstream a minimum promoter. The sentence below was added to the discussion section of the manuscript. (lines 322-325, p15)

“Our limited experimental conditions, including a small subset of cell types and reporter assays using a heterologous promoter may explain why we failed to identify enhancer functions associated with this region”.

Also, the vector where luciferase is driven by SV40 promoter and enhancer sequences (control vector) doesn't contain any insert upstream of the promoter. This SV40 control vector was only used as a positive control of the transfection. We revised this information at the methods section of the manuscript to clarify (456-457, p21)

39. The study utilizes a wide variety of tissue types, which rather confuses the overall story. If the effect of rs1888909 is really being driven by altered IL33 expression in airway epithelium, why use transgenic models where the read outs are endothelium or in the mid brain? Obviously zebrafish don't have true lungs but an epithelial read out would seem to be most relevant for a gene product which is an alarmin and given the airway epithelial cell data presented.

This choice of assay was motivated by the presence of CTCF sites and cohesin binding in the 5kb region in virtually every cell line assayed by the ENCODE Consortium, consistent with properties of sequences that have enhancer blocking function. The zebrafish assay provides an in vivo platform to test this specific property, in a system where the transgene is integrated in the genome and properly chromatinized (unlike most reporter assays which test the effects of episomal vectors in cells). We feel that this assay complements the in vitro assay in mammalian cells which show similar results, but without proper genomic context.

In the revised manuscript we were careful to clearly delineate that the information obtained from this assay simply allows us to identify regulatory differences between the risk and non-risk constructs used (See also response to question 29-Reviewer 3).

40. Given the eQTL or other data (Hi-C, TSS/enhancer mapping) suggesting ERMP1 or UHRF2 are probable targets for this variant (see Open Targets), it would be useful to add analyses looking at these genes to the ex vivo datasets.

As suggested by the reviewer, we tested rs1888909 eQTL potential for other genes at the locus in nasal epithelial cells. SNP rs1888909 was only an eQTL for the *IL33* gene. This data is shown in the table below and it was added to supplementary material (Supplementary Fig. 6).

Gene	pval	beta
ERMP1	0.13	0.049919
KIAA2026	0.14	0.052357
MLANA	0.82	-0.04081
RANBP6	0.41	-0.03995
IL33	6.14E-06	-0.24938
UHRF2	0.90	-0.00745
GLDC	0.68	0.047414

REVIEWERS' COMMENTS

Reviewer #1 (Remarks to the Author):

The authors have addressed some of my comments, and although they haven't made any changes to address my comments regarding asthma or inflammation models I take their point that this is the focus of another study, and that there is sufficient data in this manuscript. I would think that this is now of interest to a more general audience.

Reviewer #2 (Remarks to the Author):

The authors have nicely addressed all of my comments.

Reviewer #3 (Remarks to the Author):

Updated manuscript from Aneas et al respond to most if not all the comments raised by the 4 reviewers.

The work is strengthened by the addition of new experiments and gained in clarity with the removal of the Mice 4C data.

I have no further remarks on this extensive work performed by the authors. Praises for these very throughout and focused responses and modified work that is now suited for publication in Nature Com.

Reviewer #4 (Remarks to the Author):

The authors have undertaken some additional analyses to address most of the comments raised in my review of this paper, and have rewritten relevant parts of the manuscript. Whilst this still remains a complex paper to follow, it does as previously pointed out contain useful new information on this genetic association signal.